# VLM in a flash:
# I/O-Efficient Sparsification of Vision-Language Model via NEURON CHUNKING

**Kichang Yang**[*]
Seoul National University
kichang96@snu.ac.kr

**Seonjun Kim**[*]
Seoul National University
cyanide17@snu.ac.kr

**Minjae Kim**
Seoul National University
aingo03304@snu.ac.kr

**Nairan Zhang**
Meta
nairanzhang@meta.com

**Chi Zhang**
Amazon
zhanbchi@amazon.com

**Youngki Lee**[†]
Seoul National University
youngkilee@snu.ac.kr

## Abstract

Edge deployment of large Vision-Language Models (VLMs) increasingly relies on flash-based weight offloading, where activation sparsification is used to reduce I/O overhead. However, conventional sparsification remains model-centric, selecting neurons solely by activation magnitude and neglecting how access patterns influence flash performance. We present NEURON CHUNKING, an I/O-efficient sparsification strategy that operates on *chunks*—groups of contiguous neurons in memory—and couples neuron importance with storage access cost. The method models I/O latency through a lightweight abstraction of access contiguity and selects chunks with high utility, defined as neuron importance normalized by estimated latency. By aligning sparsification decisions with the underlying storage behavior, Neuron Chunking improves I/O efficiency by up to 4.65× and 5.76× on Jetson Orin Nano and Jetson AGX Orin, respectively. The code is available at https://github.com/snuhcs/vlm-flash.

## 1 Introduction

Recent vision–language models (VLMs) demonstrate strong multimodal reasoning and real-time language interaction with visual scenes. Deploying these models on edge devices is becoming essential for applications such as augmented reality (AR) and autonomous robotics that require on-device inference for robustness to limited connectivity and privacy [5, 53]. These systems must process video frames continuously without frame drops while maintaining interactive latency.

The scalability of on-device inference is fundamentally constrained by memory capacity. Edge platforms provide far less memory than what modern VLMs require. Jetson Orin Nano, for example, offers only 8 GB of memory, while LLaVA-OneVision-7B [18] requires 16 GB (fp16) for weights alone. Recent systems [2, 3, 10, 36, 49] and inference engines [1, 9] address this mismatch through weight offloading, which stores model parameters in external flash memory and loads them on demand during inference. This method allows large models to execute on small devices but introduces substantial I/O latency, which often dominates total inference time.

Activation sparsification has been widely explored to mitigate this latency [2, 44, 49]. The approach loads only the weights corresponding to neurons with high importance (e.g., magnitude of activation

---

[*]Equal contribution

[†]Corresponding author

39th Conference on Neural Information Processing Systems (NeurIPS 2025).

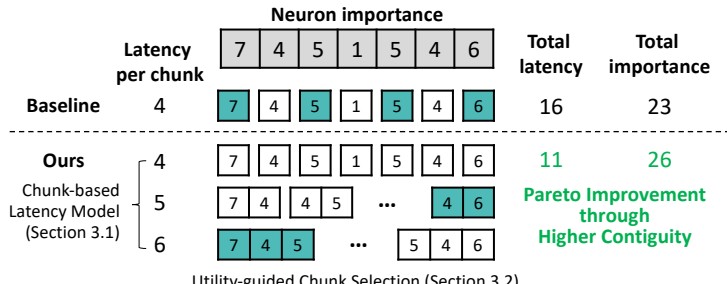

Figure 1: **Illustration of conventional sparsification *vs.* our approach**. Existing methods select neurons solely based on activation importance, which often leads to scattered, irregular access patterns with poor I/O efficiency. In contrast, our method explicitly accounts for actual I/O latency, favoring contiguous chunks that achieve better importance–latency trade-offs.

value), reducing total data transfer and improving input adaptivity. Despite its effectiveness, existing sparsification remains model-centric. It selects channels solely based on activation importance while assuming that I/O latency scales linearly with data size. Flash storage does not follow this assumption; its latency depends strongly on access contiguity, and scattered reads severely degrade throughput. Earlier methods were less affected because they targeted highly sparse LLMs or GPU memory transfers, where locality plays a smaller role. Modern VLMs exhibit smoother activation distributions and lower sparsity (Figure 2), leading to fragmented reads and high I/O overhead (Figure 4b).

We propose NEURON CHUNKING, a sparsification framework that improves the I/O efficiency of flash-offloaded inference by coupling activation sparsity with storage access behavior. The key idea is to jointly optimize neuron importance and flash I/O latency by selecting contiguous channel groups that provide a better trade-off between accuracy and latency. Contiguous reads provide higher flash throughput, allowing moderately important neighboring channels to be loaded more efficiently than distant but highly important ones (Figure 1). This design forms compact chunks that enhance access locality, leading to significant performance gains during VLM inference.

Efficient realization of this strategy requires capturing hardware I/O behavior in a form that the runtime can readily exploit. NEURON CHUNKING achieves this by reducing complex access patterns into a compact structural representation, termed the *contiguity distribution*. It summarizes how memory accesses cluster into contiguous groups (i.e., chunks) while omitting their exact spatial layout. This abstraction underlies two key components of our system. A *chunk-based latency model* profiles load latency for each chunk size and efficiently estimates the I/O latency of arbitrary access patterns from their contiguity distributions. A *utility-guided chunk selection algorithm* formulates neuron selection as a constrained optimization problem and iteratively selects chunks that maximize the importance–latency utility. Evaluation on Jetson Orin AGX and Nano using open-source VLMs and standard benchmarks shows consistent improvement in the accuracy–latency trade-off. At comparable accuracy, NEURON CHUNKING reduces I/O latency by an average of 2.19× on Nano and 2.89× on AGX, with maximum gains of 4.65× and 5.76×, respectively.

Our contributions are summarized as follows.

- We identify and characterize the hardware inefficiencies that arise when conventional activation sparsification techniques are applied to flash-offloaded VLM inference, revealing their mismatch with underlying storage access patterns.

- We propose NEURON CHUNKING, a sparsification framework that enhances flash I/O efficiency by coupling activation sparsity with storage access patterns. It jointly optimizes neuron importance and flash latency by selecting contiguous channel groups that balance accuracy and latency.

- We introduce the contiguity distribution as a compact representation of flash access behavior. Building on this abstraction, we develop a *chunk-based latency model* and a *utility-guided chunk selection algorithm* that together enable latency-aware sparsification by balancing model quality and I/O efficiency.

- We evaluate NEURON CHUNKING on Jetson Orin AGX and Nano with open-source VLMs and standard benchmarks. Results show consistent improvement in the accuracy–latency trade-off,

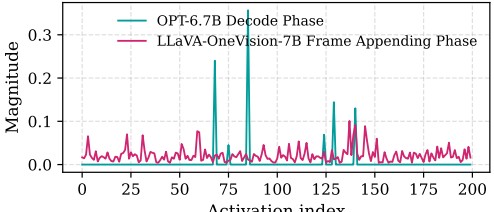

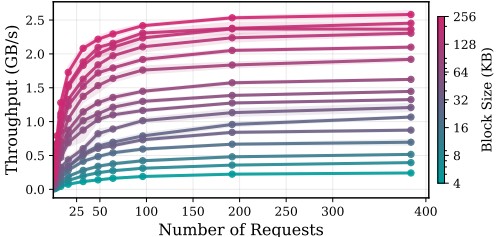

Figure 2: Activation-magnitude plot for two workloads: (teal) a ReLU-based LLM in the decode phase and (magenta) a gated-activation-based VLM in the frame appending phase. VLM exhibits a smoother distribution, with much less variation between high and low activation values.

Figure 3: Read throughput as a function of block size and number of requests, profiled on Jetson AGX Orin with a Samsung 990 Pro SSD. Throughput quickly saturates and remains stable once the request count exceeds minimal thresholds.

reducing I/O latency by up to 5.76× on AGX and 4.65× on Nano while maintaining comparable accuracy.

## 2    Background and Motivation

### 2.1    Overview of LLM/VLM Inference Pipeline

An LLM inference pipeline consists of two stages: (i) the prefill stage, where the input prompt, consisting of multiple tokens, is processed to generate key-value (KV) cache, and (ii) the decoding stage, where the model generates tokens one at a time autoregressively using KV cache.

When processing an online video stream with VLMs, an additional *frame-appending* stage is introduced between the prefill and decoding stages. In this stage, incoming video frames are processed sequentially as they arrive. VLMs integrate a vision encoder alongside the backbone LLM. Each frame is divided into patches and passed through the vision encoder, which converts them into a sequence of visual tokens. These tokens are then fed into the language model, augmenting the existing KV cache generated from the language prompt (see Appendix B.1 for details).

### 2.2    Model-side Observation: Smooth Activation Profiles in VLMs

Modern VLMs exhibit smooth activation distributions as shown in Figure 2. This smoothness is a general property of the architecture rather than a model-specific artifact. It arises from two key factors: (i) gated activation functions such as SwiGLU [35] and GeLU [12] produce continuous rather than sparse activation values, and (ii) averaging these values over multiple visual tokens, as in LLaVA-OneVision with $14 \times 14$ tokens per frame, further reduces the variation in importance scores. As shown in Appendix C, this phenomenon consistently appears across diverse models.

This observation suggests that when activation values are less distinct, selection policies can balance model quality with system performance rather than focusing solely on the largest activations. Furthermore, the lack of sharply separated activations makes it difficult to depend on only a few dominant neurons, making moderate sparsity a more favorable operating point.

### 2.3    System-side Observation: Flash I/O Sensitivity to Access Contiguity

Flash read performance depends on memory access patterns, and access contiguity is the dominant factor. As shown in Figure 4a, larger contiguous reads improve throughput until reaching the bandwidth limit, marking a shift from overhead-bound to bandwidth-bound operation. This behavior reveals a counterintuitive aspect of sparsification: while higher sparsity reduces data transfer, it fragments memory accesses and can increase latency (Figure 4b).

Notably, throughput stabilizes once the request count exceeds a minimal threshold (Figure 3). This stability enables latency estimation from access contiguity using a one-time throughput profile across block sizes, making latency-aware sparsification practical during inference.

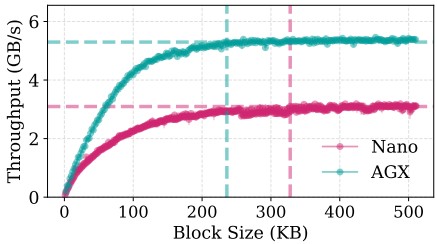

(a) Block Size vs. Flash Read Throughput

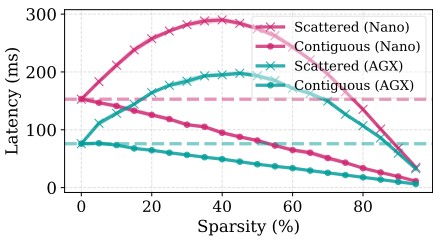

(b) Sparsity vs. Flash Read Latency

Figure 4: Flash read performance under varying access patterns. Left: Throughput vs. block size when reading 128 MB (MLP weight sizes in Qwen2-7B [50]). Right: Latency vs. sparsity across two access modes—*scattered* (random) and *contiguous* (sufficiently block-aligned to saturate throughput: 328 KB on AGX, 236 KB on Nano). Error bars show $\pm 1$ std; dashed lines indicate saturate throughput and full-load latency. Experiments use Linux direct I/O [23] with 6-thread thread-pool in C++.

## 3 NEURON CHUNKING

Motivated by the aforementioned considerations, we introduce NEURON CHUNKING, an I/O efficiency-aware activation sparsification method. Unlike traditional sparsification methods, NEURON CHUNKING jointly considers neuron importance and the memory access costs associated with retrieving selected neurons from flash memory.

To enable such a joint optimization, two key requirements must be satisfied: (i) latency estimation for a given memory access pattern, and (ii) neuron selection that balances costs and benefits. Both tasks must be executed frequently at runtime, once per weight matrix (e.g., approximately 200 times per frame for LLaVA-OneVision-Qwen2-7B [18]). Each must therefore complete within a very short time frame (i.e., about a few milliseconds). Previous approaches avoided this complexity by assuming that latency scales monotonically with the number of selected neurons—an assumption that does not hold in our target workloads.

Our key idea is to abstract a memory access pattern into a concise representation called the *contiguity distribution*, defined as the frequency distribution of the contiguity of the selected neurons. Concretely, we group consecutively selected neurons into *chunks*, each representing a maximal contiguous range of neuron indices within the selected set. For example, selecting neurons with indices $\{1, 2, 4, 6, 7\}$ yields three chunks: $\{1, 2\}$, $\{4\}$, and $\{6, 7\}$. We then model the total read latency as a function of this contiguity distribution—e.g., in the above case, one chunk of size 1 and two of size 2—while discarding global structural cues such as the exact spatial arrangement of chunks. This abstraction both simplifies hardware modeling and substantially reduces the search space of the neuron selection.

- In Section 4.1, we present a *chunk-based latency model*. It builds a lookup table of per-chunk-size latencies via offline profiling and estimates total latency directly from the contiguity distribution.

- In Section 4.2, we introduce a *utility-guided chunk selection* algorithm. Given a list of activation importance, this algorithm generates candidate neuron chunks of varying sizes and greedily selects the chunks with high utility (i.e., importance-per-latency ratio).

- In Section 4.3, we apply a lightweight offline reordering step that groups neurons based on activation statistics to improve I/O contiguity, and find that this simple approach suffices without relying on co-activation information.

### 3.1 Chunk-based Latency Model

We first build a lightweight latency estimation model that predicts the I/O latency of arbitrary neuron-access patterns on flash storage. Even after abstracting an access pattern into a contiguity distribution, the space of possible distributions remains combinatorially large (the number of combinations grows exponentially with the number of channels), making exhaustive profiling infeasible. Meanwhile, prior SSD modeling frameworks [13, 43] provide device-level fidelity but require low-level hardware configurations or full-system simulations, which are impractical during inference.

To address this, we propose a simple and scalable latency model tailored for inference-time sparsification. We approximate the total read latency of the access pattern as the sum of the latencies of its constituent chunks. Let the selected chunks be $C_i$ $(i = 1, \ldots, n)$, each of size $s_i$. The total latency is estimated as $L_{\text{total}} = \sum_{i=1}^{n} T[s_i]$ where $T[s]$ denotes the profiled read latency for a chunk of size $s$.

**Profiling $T[s]$.** We build a lookup table for $T[s]$ via offline microbenchmarks: for each chunk size $s$, we place a throughput-saturating number of chunks of size $s$ at fixed strides and measure steady-state read latency (See Appendix D for details). Fixed overheads such as command setup or metadata access during flash read initiation are amortized and become negligible in $T[s]$. This procedure yields stable per-size latencies with low measurement variance.

**Empirical Validation.** We evaluate the effectiveness of our latency model across different devices and models, as shown in Figure 5. Each plot shows actual and estimated latency values of loading selected chunks from our selection algorithm (Section 3.2).

We observe a near-linear relation between estimated and measured latency, indicating a consistent proportional bias.[3] This bias arises from our contiguity-distribution abstraction and profile setup. The lookup table is built under idealized conditions, where each chunk size is measured in isolation with uniform strides. In contrast, real access patterns interleave diverse chunk sizes and strides, invoking pattern-dependent controller and queue behaviors. These effects accumulate and average out to a proportional lift in actual latency. The near-linear correlation weakens for smaller models or lower-end devices, where I/O concurrency is lower and controller dynamics amplify tail latency, reducing the averaging effect. Importantly, the error remains near-linear, leaving the greedy chunk selection algorithm unaffected (see Section 3.2).

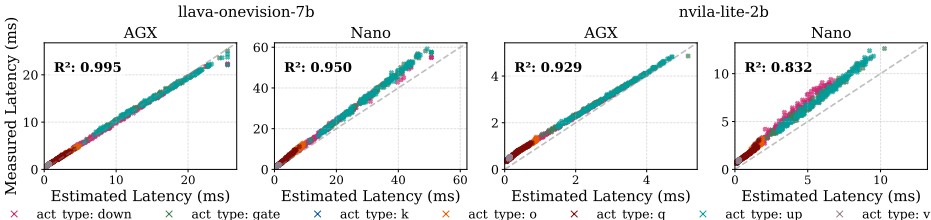

Figure 5: Comparison between real and estimated flash access latency across models and devices.

## 3.2 Utility-Guided Chunk Selection

Given the latency model, our goal is to jointly consider latency and neuron importance when selecting neurons. The selection problem is intractable, as the number of possible neuron combinations grows exponentially with the model size. To make the problem tractable, we leverage the contiguity-distribution abstraction together with the additive latency assumption, where total latency is approximated as the sum of per-chunk latencies. Under this scheme, latency becomes decomposable across chunks, which naturally motivates a greedy, chunk-level selection strategy.

### 3.2.1 Problem Formulation

Given activation magnitudes $V \in \mathbb{R}^N$ (where N is the number of neurons) and a selection budget $R$ (the number of channels to load), we aim to output a binary selection mask $M \in \{0, 1\}^N$ that maximizes importance per latency:

$$\max_{M \in \{0,1\}^N} \frac{\sum_{i=0}^{N-1} V_i \cdot M_i}{\text{Latency}(M)} \quad \text{s.t.} \quad \sum_{i=0}^{N-1} M_i \leq R,$$

where $\text{Latency}(M)$ is the estimated cost of loading the selected channels, modeled via the contiguity distribution of $M$.

---

[3]The latency gap between two points that differ by one additional chunk remains roughly proportional across the entire plot.

### 3.2.2 Algorithmic Procedure

Our algorithm consists of three main stages:

**1. Candidate chunk generation.** We construct candidate chunks by sliding windows of multiple sizes over the linear space of neuron indices, and each window position yields one candidate chunk. The maximum chunk size is set to the hardware-specific point where throughput saturates (Section 2.3). The minimum chunk size and the stride between windows are tunable hyperparameters that control the trade-off between search granularity and computational overhead of the algorithm.

**2. Chunk evaluation.** Each chunk is assigned a utility score, defined as the sum of its neuron importance values divided by its estimated latency. Latency is obtained efficiently from a pre-profiled lookup table based on our latency model.

**3. Greedy selection.** Chunks are sorted by utility, and the algorithm iteratively selects the highest-ranked ones while excluding those that overlap with previously selected chunks, continuing until the selection budget is met.

Because the latency model error is approximately linear, it merely scales all utility scores by a constant factor without changing their relative order. Therefore, the output of the greedy selection remains unaffected.

One limitation of this approach is that it does not account for potential synergies among adjacent low-scoring chunks, which could collectively form a more latency-efficient region. Even so, we observe that the algorithm performs robustly in practice and effectively identifies high-importance subsets within its runtime constraints (see Section 4.2).

A complete implementation, including mask updates, stride scheduling, and latency lookup logic, is provided in Appendix E.

### 3.3 Additional Optimization: Hot-cold Reordering

Previous works [2, 44] have observed neuron co-activation patterns and improved I/O efficiency in ReLU-based LLMs through offline reordering based on these statistics. Motivated by these findings, we explore a simpler reordering scheme that leverages hot–cold activation patterns observed in prior studies [38, 49]. Specifically, we reorder neurons according to their activation frequency, which yields comparable I/O efficiency improvements to co-activation-based methods without the need for complex optimization (See Appendix F, G for details). Thus, we adopt this hot–cold reordering as a preprocessing step during the offline profiling stage.

The procedure is as follows. We first count how frequently each neuron is activated (designating the top 50% by importance as active) using a calibration dataset. Then we sort the neurons in decreasing order of activation frequency. Based on this ordering, we permute the corresponding rows of the weight matrix so that frequently activated neurons are placed together. At runtime, the same permutation is applied to the activation vector, aligning it with the reordered weights. The runtime permutation operation incurs negligible overhead. For example, profiling the down-projection layer of LLaVA-OneVision-7B on Jetson Orin Nano over 100 trials showed a mean overhead of 1.5 ms with a 95% confidence bound of 1.8 ms, representing less than 0.02% of total inference latency.

## 4 Evaluation

### 4.1 Experimental Setup

**Hardware.** All experiments were performed on two different embedded device setups, representing low-end and high-end hardware environments:

- Jetson Orin Nano (8 GB memory) with SK Hynix Gold P31 SSD (peak sequential read: 3500 MB/s)

- Jetson Orin AGX (32 GB memory) with Samsung 990 Pro SSD (peak equential read: 7450 MB/s)

We cache the vision encoder and KV cache in memory. All weights of the backbone LLM are loaded from flash on demand. Unless specified, we use Jetson Orin Nano as our default device, reflecting the memory-constrained setting.

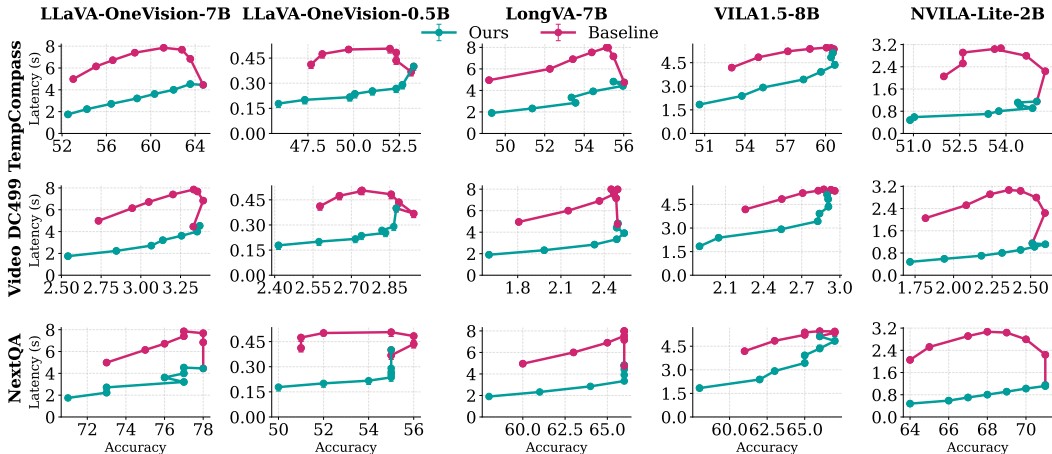

Figure 6: End-to-end performance on Jetson Orin Nano. Latency reported with error bar $\pm 1$ std.

**Models and Datasets.** We use VLM models that operate frame-by-frame, as models that process entire videos at once do not fit our real-time streaming scenario. We evaluate five models with various sizes: LLaVA-OneVision-Qwen2-7B, LLaVA-OneVision-Qwen2-0.5B [18], Llama-3-VILA1.5-8B [22], NVILA-Lite-2B [27], and LongVA-7B [54]. For datasets, we use two multiple-choice video QA benchmarks—TempCompass [25] and NExT-QA [47] (randomly sampled 3000 examples)—and a video description dataset, VideoDetailCaption [29]. Unless otherwise specified, we use LLaVA-OneVision-Qwen2-7B and the TempCompass dataset by default.

**Comparison Setup.** As a baseline, we implement top-$k$ activation sparsification that selects the most important channels based on activation magnitude, following prior works [2, 16, 24]. We apply TEAL's [24] profiling-based method to determine layer-wise sparsity levels for both the baseline and our method, using 25 out of 410 videos from the TempCompass dataset (excluded from the main evaluation). See Appendix H for hyperparameter details.

**Metric.** We report accuracy and I/O latency. Accuracy is defined as the ratio of correct answers on multiple-choice QA datasets, and as a 0–5 score from ChatGPT-based evaluation on video description dataset, following prior works.[4] Due to the large dataset size, accuracy is measured using a Supermicro A+ Server 4124GS-TNR with 8 RTX A6000 GPUs. We measure accuracy under a sparsity level from 0% to 70% in 10% increments. Each latency experiment was repeated 30 times under identical conditions. We report the median latency together with 95% confidence intervals computed by a 10000-sample non-parametric bootstrap (bias-corrected and accelerated, BCa). We enable `jetson_clocks` and disable swap to ensure stable measurement.

### 4.2 Experimental Results

**Accuracy-Latency Trade-off.** Figure 6 shows the accuracy–latency curves for the baseline and our method across all models and datasets. Our method consistently achieves a better accuracy–latency trade-off, with an average I/O speedup of 2.19× and up to 4.65× at comparable accuracy levels based on linear interpolation. The baseline often suffers from poor latency, especially at low to medium sparsity levels, occasionally increasing total latency—consistent with our observations in Figure 4. This issue is pronounced in smaller models, where channels are smaller, leading to more fragmented I/O. In contrast, our method decisively addresses these inefficiencies by tailoring selection to storage behavior, enabling consistently faster inference. Note that the slight accuracy gain at higher sparsity can appear when weak or noisy activations are removed. Similar regularization effects have been reported in pruning-based model compression work [11, 15].

**Cross-Device Evaluation.** Figure 7 presents results on Jetson Orin AGX (the full results provided in Appendix I show consistent trends). Our method delivers similar relative improvements, achieving

---

[4]For VideoDetailCaption, we queried the OpenAI endpoint `gpt-4o-mini-2024-07-18`, a more recent and stronger version than used in earlier work; thus, results are not directly comparable.

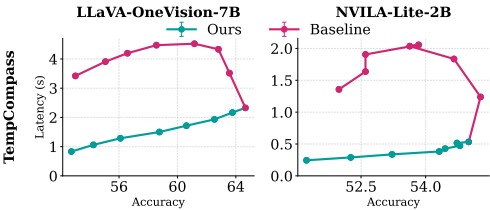

Figure 7: End-to-end performance on AGX.

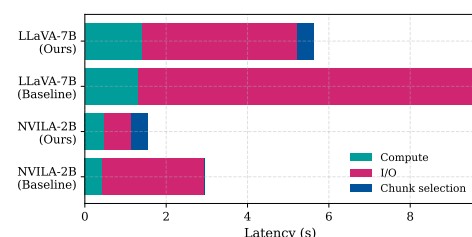

Figure 8: Latency breakdown.

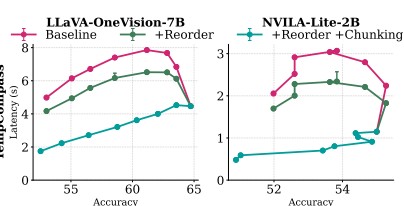

Figure 9: Ablation of two components: Reordering and Chunk selection.

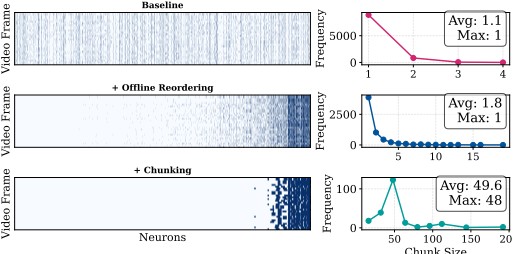

Figure 10: Contiguity distribution before/after our techniques are applied.

an average 2.89× speedup and up to 5.76× I/O latency reduction, computed over the full set of models and datasets. The larger speedup on AGX reflects its wider throughput gap between contiguous and scattered access.

**Latency Breakdown.** Figure 8 shows the latency breakdown at a 5% accuracy drop. The end-to-end speedup is smaller than the I/O-only gain because compute time remains nearly constant, so its relative share increases as I/O time decreases. This gap could narrow with optimized kernels or I/O–compute overlap [6, 10], which we do not apply in our evaluation. Our method substantially reduces I/O latency while incurring a slight compute increase, as maintaining the same accuracy requires loading marginally more channels. Nevertheless, the overall latency still decreases because data is fetched contiguously rather than through scattered accesses. The chunk selection overhead is modest—about 2 ms per weight matrix, totaling roughly 400 ms for the full model—which is small relative to total inference time.

**Ablation Study.** Figure 9 shows the accuracy–latency trade-off as each component is incrementally added: baseline, with hot–cold reordering, and with both reordering and chunk selection. For the LLaVA-7B model, hot–cold reordering yields up to a 1.23× speedup, which increases to up to 2.55× when chunk selection is additionally applied. Notably, online chunk selection plays a critical role, as the optimal subset of neurons is input-dependent and cannot be determined offline.

**Visualization of Utility-Guided Chunk Selection** Figure 10 visualizes selected channels for three variants—baseline, baseline with reordering, and baseline with both reordering and chunk selection—along with the contiguity distributions at matched accuracy. Reordering yields only modest gains by loosely clustering frequently activated channels. In contrast, chunk-based selection drives the dominant improvement: it targets high-utility contiguous regions, raising the average chunk size from roughly 1–2 to nearly 50. See Appendix J for results across a broader range of settings.

## 5 Discussion and Future Work

**Generalization to other models and workloads.** The proposed framework extends beyond vision-language models to a broader class of architectures and inference settings. The same principle of hardware-aware structured sparsification applies naturally to multi-token LLM inference scenarios such as speculative decoding, parallel sampling, and batched interactive serving, where activations aggregated across tokens yield smoother neuron-importance distributions. This property enables

latency-aware sparsification to maintain responsiveness in real-time, user-facing applications including chat assistants and copilots that operate under tight latency constraints.

The approach also generalizes to plain LLMs and ViT-based models that exhibit smooth activation magnitudes and operate under I/O-bound conditions. Recent LLMs increasingly employ non-ReLU activations such as SwiGLU or GeLU, making them amenable to our chunking formulation. Similarly, ViT-based models on edge devices benefit from reduced access fragmentation across smaller channel dimensions. Overall, these characteristics indicate that the proposed framework provides a general foundation for coupling structured sparsity with hardware-aware optimization across diverse model families. Additional details and preliminary results on LLMs and ViTs are presented in Appendix N.

**Impact of Emerging I/O Mechanisms.** Emerging I/O frameworks such as io_uring [4] offer improved support for asynchronous and scattered reads, which may reduce the performance gap between random and contiguous access. However, advances in storage hardware (e.g., internal prefetching, read coalescing) will likely continue to favor contiguity, suggesting that structured access optimization will remain beneficial.

**Leveraging Additional Memory Budget for Caching.** While our method assumes minimal memory availability, additional latency reduction is possible when the device has sufficient memory to cache frequently accessed weights. Caching strategies (e.g., hot-neuron caching) proposed in prior works [2, 38, 49] can be applied in a complementary manner by simply assigning zero importance to cached neurons. Once hot weights are cached, the remaining uncached accesses become more scattered (even after reordering), making our chunk-based selection more critical for sustaining I/O efficiency. However, if the device has sufficient memory to cache a large portion of the model weights, the overall flash I/O volume becomes negligible, reducing the benefit of our method.

## 6 Related Work

### 6.1 Activation Sparsification

Activation sparsification, which selectively loads weight channels corresponding to large activations, has been widely studied (See Appendix B.2 for details). Deja Vu [28] observed that MLP layers in LLMs exhibit significant dynamic sparsity, while CATS [16] extended this insight to modern LLMs, which utilize gated MLPs with non-ReLU activations. TEAL [24] further explored sparsification by applying it to attention layers. However, these methods are model-centric—they sparsify solely based on activation magnitude without considering hardware-level access patterns. This design choice was reasonable in their settings, where all weights reside in GPU VRAM and the bandwidth between device and shared memory saturates quickly even with limited access contiguity. In contrast, in flash-offloaded settings, such model-centric sparsification leads to significant performance degradation. Other approaches [31, 37, 39] seek to apply ReLU-ification to non-ReLU-based LLMs, fine-tuning them to enhance sparsity. Although effective, this method demands extensive retraining on a minimum of 50 billion tokens.

### 6.2 LLM Weight Offloading

LLM weights often exceed GPU VRAM capacity, prompting various offloading strategies. Some approaches [3, 36, 38] offload weights to CPU memory. This is impractical on edge SoCs with unified memory, where the CPU and GPU draw from the same DRAM pool and no additional capacity is gained [32].

Several recent works [2, 44, 49] adopt flash-based offloading and propose techniques to reduce I/O latency. LLM in a Flash [2] and PowerInfer-2 [49] improve I/O efficiency by bundling channels across projection layers, but the resulting gains in access contiguity are limited and rely on large memory budgets for caching (see Appendix L). Ripple [44] enhances access locality through offline neuron reordering, yet its reliance on ReLU-based sparsity and lack of hardware-aware runtime mechanisms limit its effectiveness to modern VLMs (see Section 4.2). These approaches were sufficiently effective for ReLU-based LLMs with high activation sparsity, where the total I/O volume was low enough to offset efficiency degradation. In contrast, VLMs exhibit smoother activation distributions and higher I/O demand, where such techniques fail to sustain efficiency under realistic I/O constraints.

### 6.3 LLM Compression

Various LLM compression techniques, including quantization [7, 21, 46], weight pruning [14, 26, 30, 48], and distillation [33, 40, 41], have been proposed to reduce computational and memory overhead. In contrast to these static compression approaches, activation sparsification is inherently input-adaptive, providing a distinct advantage in exploiting runtime activation dynamics. These methods are orthogonal to activation sparsification and can be combined to effectively mitigate I/O latency from storage devices [24].

## 7 Conclusion

We presented NEURON CHUNKING, a latency-aware activation sparsification approach tailored for flash-offloaded VLM inference. Unlike prior methods that treat I/O latency as a function of volume alone, our method models the performance implications of access contiguity and aligns neuron selection with storage behavior. We show that our contiguity-based latency model and utility-guided chunk selection algorithm consistently improve the accuracy–latency trade-off. These results underscore the importance of co-designing sparsification with hardware characteristics for efficient edge inference.

## Acknowledgments and Disclosure of Funding

We sincerely thank our anonymous reviewers for their valuable comments. This work was supported by National Research Foundation (NRF) funded by the Korean government (MSIT) (No. RS-2024-00463802).

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

# A   Appendix Overview

This appendix provides additional details supporting the main paper.

**Default Setup.**   Unless otherwise specified, we adopt the same default configuration as used in the main paper's evaluation: Llava-OneVision-Qwen2-7B [18] as the model and the TempCompass [25] dataset as the input source. For end-to-end evaluation on Jetson Orin AGX (Appendix I), we use the full set of evaluation datasets following the setup in the main paper. For analysis-oriented experiments—including visualization, reordering, and activation statistics—we use a subset of 25 videos from TempCompass that were excluded from the main evaluation (as described in Section 5.1). In the reordering experiment, 20 videos are used for calibration and 5 for validation (i.e., result visualization). When analyzing attention and MLP modules, we focus on the q, o, gate, and down projections, and omit k, v, and up since they share input activations with q and gate, respectively. Additionally, we include three representative layers—early (0), middle (13), and late (27)—to capture variation across layer depths (LLaVA-OneVision-Qwen2-7B has 28 layers). We restate this setup where relevant throughout the appendix.

# B   Extended Background on VLM Sparsification

## B.1   Vision–Language Model Inference Process

A vision–language model consists of a vision encoder $f_{vision}$, a projector $f_{proj}$, and a backbone LLM $f_{llm}$.

Given a language prompt $p = \{t_1, t_2, \ldots, t_m\}$ of $m$ tokens, the model first performs a *prefill* step that processes all tokens at once to generate key–value (KV) caches for each transformer layer:

$$t_{m+1}, \ \mathrm{KV}_{<m+1} = f_{llm}(p)$$

In the *frame appending* stage, each video frame $F_i$ ($i = 1, \ldots, N$) is processed upon arrival. Each frame is encoded into $n$ visual tokens via the vision encoder and projector:

$$v^{(i)} = \{v_1^{(i)}, v_2^{(i)}, \ldots, v_n^{(i)}\} = f_{proj}(f_{vision}(F_i))$$

These tokens are then fed into the LLM to produce additional KV pairs, which are appended to the existing cache:

$$t_{m+ni+1}, \ \mathrm{KV}_{<m+ni+1} = f_{llm}(v^{(i)}, \mathrm{KV}_{<m+n(i-1)+1})$$

During the prefill and frame appending stage, the token output of $f_{llm}$ is ignored; only the KV cache is used in subsequent decoding.

In the *decoding* stage, a new token is generated one at a time autoregressively. At decoding step $j$, the model takes the previous token $t_{m+nN+j}$ and the current KV cache to generate the next token:

$$t_{m+nN+j+1}, \ \mathrm{KV}_{<m+nN+j+1} = f_{llm}(t_{m+nN+j}, \mathrm{KV}_{<m+nN+j})$$

The decoding stage may begin in one of two ways: either via an explicit query provided by the user (e.g., a natural language instruction, which is appended in a similar way to visual tokens), or by designing the model to emit a special control token at important frames, signaling that decoding should commence. In the former case, decoding starts after appending the query tokens; in the latter, the final generated token from the last frame is preserved and used as the first input of the decoding stage.

## B.2   Activation Sparsification

**Notation.**   Let the activation vector (also referred to as hidden states) for a single layer be $a \in \mathbb{R}^m$ and the corresponding weight matrix be $W \in \mathbb{R}^{m \times n}$, where $m$ is the number of neurons (also referred to as channels) and $n$ is the output dimension. Each row $W_i \in \mathbb{R}^n$ of $W$ corresponds to a single neuron, contributing to the output through the dot product $a_i W_i$. Thus, the output $y \in \mathbb{R}^n$ is computed as:

$$y = a^\top W = \sum_{i=1}^{m} a_i W_i,$$

which can be interpreted as a weighted sum over neurons, where the activation values $a_i$ act as per-sample dynamic weights.

**Saliency via Magnitude.** In modern LLMs where non-ReLU activation functions (e.g., SwiGLU, GeGLU) are standard, activation values are not exactly zero—as opposed to ReLU-based activation functions, which produce exact zeros for inactive neurons. As a result, identifying salient neurons—those most critical to output quality—is nontrivial. Prior works such as TEAL [24] and CATS [16] propose using the magnitude of activations as a proxy for saliency. This approach assumes that neurons with higher $|a_i|$ contribute more significantly to the output.

**Magnitude-Based Sparsification.** Given a sparsity target $s \in [0, 1)$, the goal is to retain only the top-$(1-s)m$ neurons per input based on their importance. The process is as follows:

1. Compute importance scores $v_i = |a_i|$ for $i = 1, \ldots, m$.
2. Select a binary mask $M \in \{0, 1\}^m$ such that $M_i = 1$ if $|a_i|$ is among the top-$(1-s)m$ entries of $v$, and $M_i = 0$ otherwise.
3. Construct the sparsified output:

$$\tilde{y} = \sum_{i=1}^{m} M_i a_i W_i.$$

This technique is input-dependent and requires re-evaluation of $M$ at each inference step. While simple and effective, it does not consider the memory access cost of retrieving weight rows from flash storage, which becomes critical in flash-offloaded inference settings. An alternative to top-$(1-s)m$ selection is to use a fixed activation threshold to filter out low-importance neurons.

In vision-language models (VLMs), where a single input (e.g., image) corresponds to multiple tokens, we extend this method by computing the importance of each neuron as the average absolute activation magnitude across tokens. This yields a single importance vector per input, allowing sparsification to proceed as in the single-token case.

## C  Additional Evidence of Activation Smoothness Across VLMs

To further validate that the smoothing effect is a general architectural property of VLMs rather than a model-specific behavior, we measured the coefficient of variation (CV) of neuron importance before the down-projection layer—where conventional sparsification is typically applied in ReLU-based LLMs—across multiple VLM architectures and a ReLU-based baseline (OPT-6.7B).

Table 1: Coefficient of variation (CV) of neuron importance before the down-projection layer across multiple models.

| Layer | LLaVA-7B | LLaVA-0.5B | VILA-8B | NVILA-2B | LongVA | OPT-6.7B |
|-------|----------|------------|---------|----------|--------|----------|
| First | 1.44     | 1.31       | 1.25    | 1.07     | 1.20   | 11.65    |
| Mid   | 1.25     | 1.33       | 1.38    | 1.32     | 1.34   | 8.63     |
| Last  | 3.30     | 3.58       | 2.48    | 4.55     | 3.01   | 9.19     |

Across all VLM models, the CV values (1.07–4.55) are dramatically lower than those of the ReLU-based baseline (8.63–11.65), demonstrating that smooth activation distributions are a consistent property of modern VLM architectures. This smoothing effect makes contiguity-aware selection particularly beneficial for VLMs: when importance differences between neurons are small, I/O efficiency becomes the decisive factor for overall performance.

## D  Benchmark Details

We profiled read throughput as a function of chunk size on two devices: Jetson AGX Orin (Samsung 990 Pro SSD) and Jetson Orin Nano (SK Hynix Gold P31 SSD). Each device reaches 99% of its peak

throughput at approximately 236 KB (AGX) and 348 KB (Nano). Measurements were taken in 1 KB increments up to the saturation point, with all runs completing within 20 minutes per device.

**Profiling Setup.**

- Prepare a large dummy file (e.g. 128MB) on flash-backed storage.

- Issue sequential reads of size $s \in \{1\,\mathrm{KB}, 2\,\mathrm{KB}, \dots, S_{\max}\}$, where $S_{\max}$ is the smallest size reaching 99% of peak throughput.

- Record average throughput over multiple trials.

Throughput variance was negligible (standard deviation $<1\%$ of the mean) across all sizes.

# E   Algorithm Implementation Details

Algorithm 1 provides a pseudocode of our multi-scale chunk selection method. Below, we describe the corresponding implementation in detail, which is designed for runtime efficiency and integrates both CPU and GPU components.

**Inputs.**   The inputs to Algorithm 1 are:

- $V \in \mathbb{R}^N$: Activation magnitudes.

- $R$: Total number of rows to select.

- `row_size_KB`: Size of each row in kilobytes, used to convert kilobyte-based parameters to row units.

- $[s_{\min}, s_{\max}]$ and $\Delta s$ (in KB): Define the chunk size range and the granularity of sizes considered.

- `jump_cap` (in KB): Limits the maximum stride between starting indices of candidate chunks for efficiency.

  – By default, stride equals the chunk size (i.e., non-overlapping).
  – If the chunk size exceeds `jump_cap`, stride is clipped to the cap, allowing overlapping candidates.

- $L(\cdot)$: Device-specific latency lookup function mapping chunk size (in rows) to access cost.

- In the actual implementation, a device flag selects the appropriate lookup table (AGX or Nano). In the pseudocode, this is simplified by directly passing $L(\cdot)$.

**Prefix Sum.**   To enable constant-time computation of the sum of importance for any contiguous chunk, a CPU-side prefix sum of the activation magnitudes is first computed.

**Chunk Candidate Generation.**   For each chunk size $s$ (converted to row count), the algorithm slides a window across the activation vector in steps of $\min(s, \texttt{jump\_cap})$. Each candidate chunk is scored using the ratio of summed importance to estimated latency, with latency values retrieved from a pre-profiled lookup table $L(s)$ based on hardware throughput (see Appendix D).

**GPU Sorting.**   The importance-to-cost scores of all candidate chunks are transferred to GPU memory and sorted in descending order using PyTorch's GPU-accelerated sort. This step enables scalable candidate prioritization with minimal overhead.

**Greedy Selection.**   Candidates are selected greedily based on the sorted scores. Each selected chunk is added to the output mask if it does not overlap with already selected rows and does not exceed the remaining budget $R$. Overlaps are checked with early termination, and the mask is updated in-place.

The algorithm design reflects the trade-off between optimality and runtime feasibility: by limiting the chunk search space and leveraging GPU sorting, it enables input-dependent sparsification at inference time within few milliseconds latency.

**Algorithm 1** Multi-scale Chunk Selection

---

**Require:** Activation magnitudes $V \in \mathbb{R}^N$, number of rows to select $R$, row size in KB, chunk size range $[s_{\min}, s_{\max}]$ in KB, step size $\Delta s$ in KB, jump cap in KB, latency lookup function $L(\cdot)$
**Ensure:** Binary mask indicating selected rows
 1: Convert chunk-related parameters to row units:
$\qquad r_{\min} \leftarrow \max(1, \lfloor s_{\min}/\text{row\_size\_KB} \rfloor)$
$\qquad r_{\max} \leftarrow \max(1, \lfloor s_{\max}/\text{row\_size\_KB} \rfloor)$
$\qquad \Delta r \leftarrow \max(1, \lfloor \Delta s/\text{row\_size\_KB} \rfloor)$
$\qquad \text{jump\_cap\_rows} \leftarrow \max(1, \lfloor \text{jump\_cap}/\text{row\_size\_KB} \rfloor)$
 2: Compute prefix sum array: $\text{cumsum}[0:N] \leftarrow \text{prefix\_sum}(V)$
 3: Initialize empty candidate list $C$
 4: **for** $r$ from $r_{\min}$ to $r_{\max}$ with step $\Delta r$ **do**
 5: $\quad$ stride $\leftarrow \min(r, \text{jump\_cap\_rows})$
 6: $\quad$ **for** $i = 0$ to $N - r$ with step stride **do**
 7: $\qquad$ benefit $\leftarrow \text{cumsum}[i + r] - \text{cumsum}[i]$
 8: $\qquad$ cost $\leftarrow L(r)$
 9: $\qquad$ Append candidate (benefit/cost, $i, r$) to $C$
10: $\quad$ **end for**
11: **end for**
12: Sort $C$ by score descendingly (GPU-accelerated)
13: Initialize mask$[0:N] \leftarrow 0$, selected $\leftarrow 0$
14: **for** candidate $(\_, i, r)$ in sorted $C$ **do**
15: $\quad$ **if** chunk overlaps selected rows or $r > R - \text{selected}$ **then**
16: $\qquad$ **continue**
17: $\quad$ **end if**
18: $\quad$ Set mask$[i : i + r] \leftarrow 1$
19: $\quad$ selected $\leftarrow$ selected $+ r$
20: $\quad$ **if** selected $\geq R$ **then**
21: $\qquad$ **break**
22: $\quad$ **end if**
23: **end for**
24: **return** mask

---

## F  Neuron Activation Frequency Analysis

Figure 11 illustrates the distribution of neuron activation frequency across different layers when the effective sparsity is 40%. The plot is structured as a $3 \times 4$ grid, where each row corresponds to a layer and each column to an activation type. Many neurons are neither always-on nor always-off, confirming the presence of input-dependent sparsity in VLMs, consistent with prior findings in LLMs [20, 28]. This suggests that input-aware sparsification remains effective in our setting, although such dynamic sparsity inevitably leads to fragmented access patterns.

Additionally, TEAL profiling introduces sparsity variation across layers, resulting in some layers with very high or low sparsity (e.g., q projection of layer 0 has 94% sparsity). These layers exhibit a high proportion of hot or cold neurons, suggesting that simple offline hot–cold reordering can be effective for improving contiguity in these cases.

## G  Impact of Offline Reordering Schemes

Although offline reordering is not our primary focus—we target online policies—Figure 12 compares the contiguity of selected neurons before and after applying offline reordering, using either hot–cold reordering or Ripple's [44] coactivation-based method. Both methods yield modest improvements over the original ordering, with comparable gains across most layers. While Ripple performs better in one case (the o projection of layer 0), the overall difference is minor, suggesting that hot–cold reordering offers a lightweight and effective alternative.

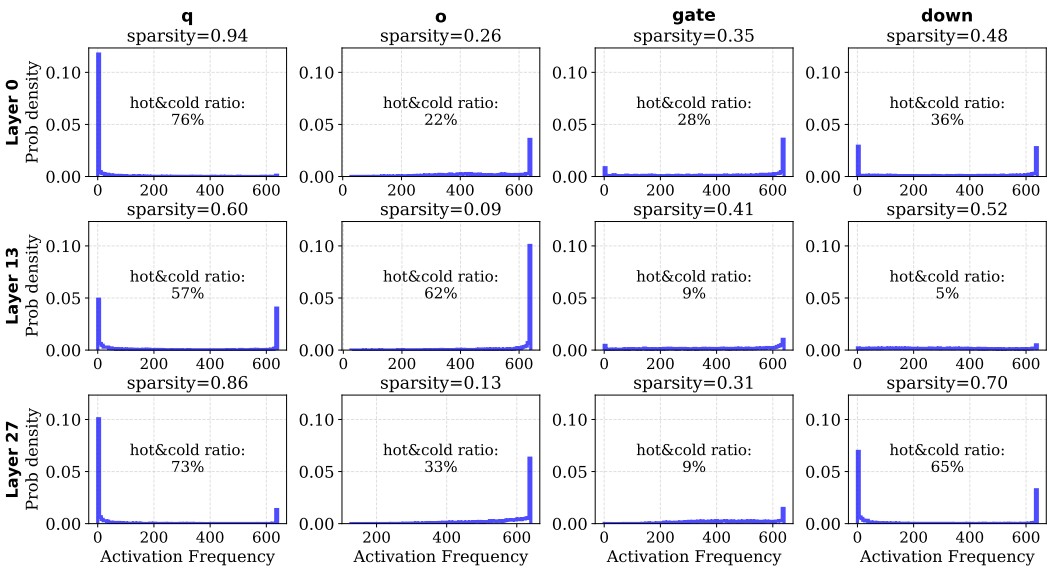

Figure 11: Activation frequency of neurons across layers, with effective sparsity set to 40% (layer-wise sparsity determined by TEAL [24] profiling). The text in the center of each plot indicates the proportion of hot neurons (activated >99% of the time) and cold neurons (<1%).

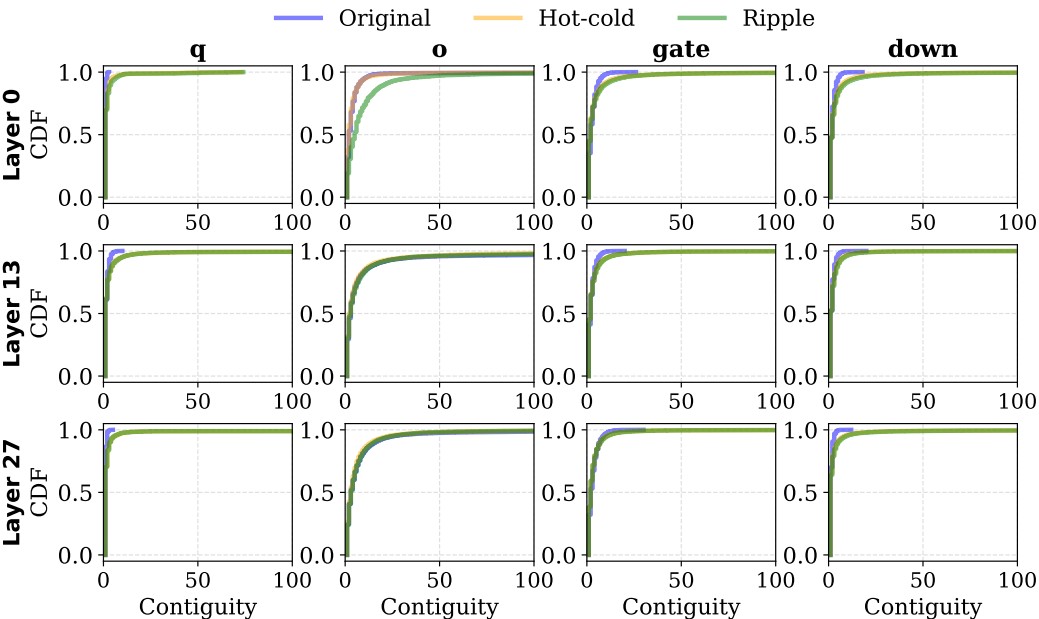

Figure 12: CDF of contiguity of selected neurons before and after reordering, with sparsity=0.4.

# H    Hyperparameter Selection

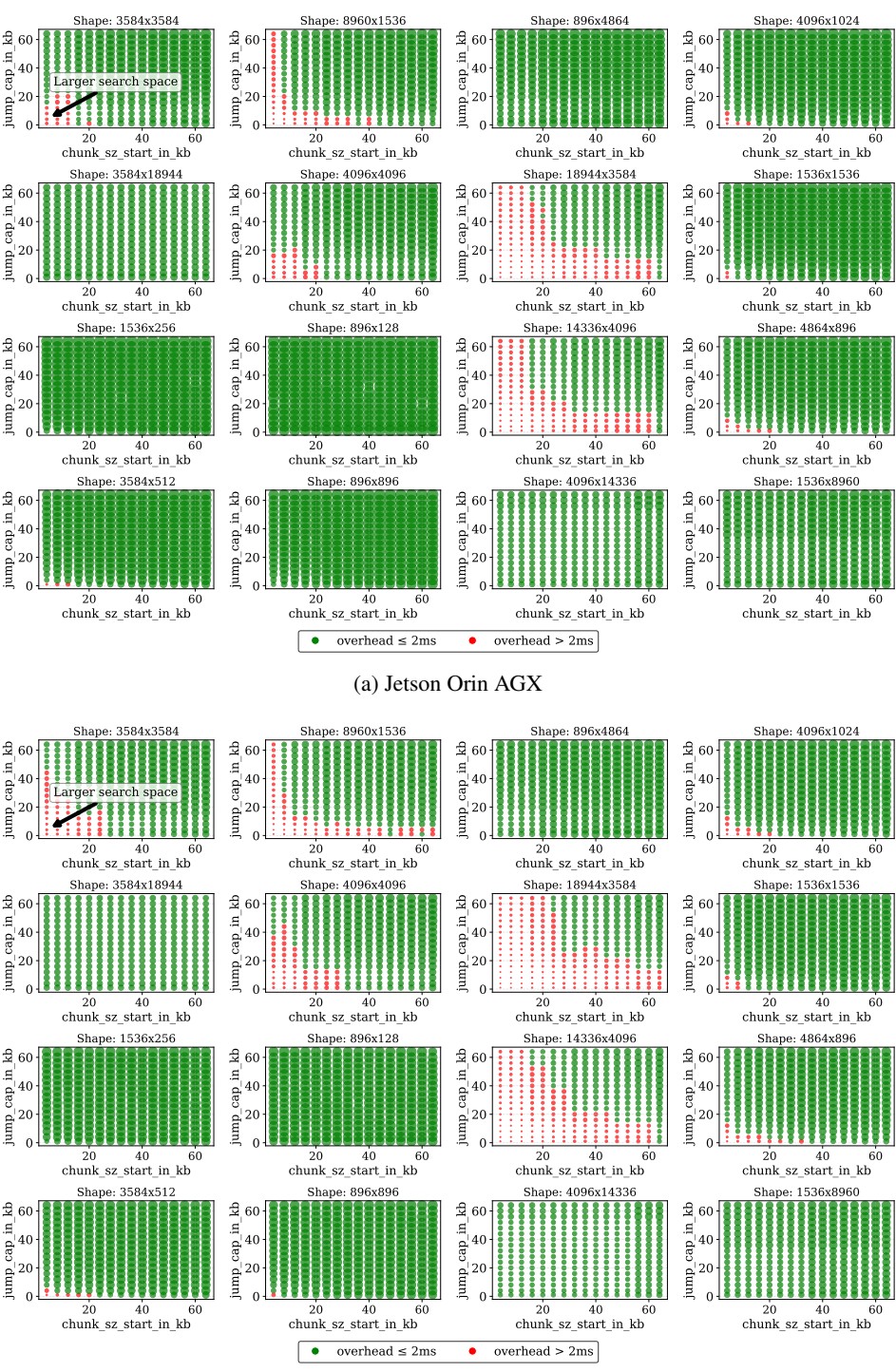

(a) Jetson Orin AGX

(b) Jetson Orin Nano

Figure 13: Runtime overhead of chunk selection across hyperparameter configurations on Jetson Orin AGX (top) and Jetson Orin Nano (bottom). Each point represents a configuration defined by starting chunk size (chunk_sz_start_in_kb, $x$-axis) and jump cap (jump_cap_in_kb, $y$-axis). Step size is set equal to the start size; the chunk size end is fixed from I/O profiling (236 KB for AGX, 348 KB for Nano). Circle size is inversely proportional to runtime, and color indicates whether the 2 ms latency threshold is exceeded.

The hyperparameters of our chunk selection algorithm are chosen with two objectives in mind: (i) the runtime overhead must remain within a practical latency threshold (under 2 ms), and (ii) the compromise in selection quality for computational efficiency should be minimal.

We adopt a two-stage selection strategy. First, we filter out configurations that exceed 2 ms of runtime overhead. Since the overhead depends on the shape of the weight matrix, we benchmark each configuration across representative matrix shapes drawn from the models used in our evaluation. Measurements are conducted at sparsity 0.1 to conservatively capture the worst-case overhead.

Among the remaining feasible configurations, we heuristically select those near the lower-left region of the search space—where chunk sizes grow in a fine-grained manner and the chunk stride is small. These settings allow for broader search coverage while maintaining overhead within budget.

Figure 13 shows the measured overhead for Jetson Orin AGX (top) and Jetson Orin Nano (bottom). For each device, we sweep over the starting chunk size (`chunk_sz_start_in_kb`, $x$-axis) and the jump cap (`jump_cap_in_kb`, $y$-axis), where the hyperparameter space spans from 0 to 64 KB in 4 KB increments. For simplicity, the step size is set equal to the start size, and the end size is fixed from I/O profiling—236 KB for AGX and 348 KB for Nano. Each configuration is evaluated 30 times using randomly generated activation magnitudes. This provides a reliable estimate, as over 80% of the total runtime is dominated by GPU sorting via a data-independent radix sort [34], allowing random inputs to be used for measuring overhead.

We observe two clear trends: (i) configurations involving large weight matrices (e.g., 18944$\times$3584) tend to incur higher overhead, making some configurations infeasible; (ii) AGX supports more configurations due to its higher compute capacity compared to Nano.

Final hyperparameters are selected near the boundary between feasible (green) and infeasible (red) regions, with a small margin for safety. The selected settings are summarized in Table 2.

While we have not conducted a full sensitivity analysis, our empirical findings suggest that the method performs consistently well across a range of hyperparameter settings. A more thorough investigation into the effects of different configurations remains an interesting direction for future work.

Table 2: Selected hyperparameters per weight matrix shape on Jetson Orin AGX and Nano

| Shape (Rows $\times$ Cols) | AGX | | Nano | |
|---|---|---|---|---|
| | chunk_sz | jump_cap | chunk_sz | jump_cap |
| (3584, 3584) | 20 | 20 | 24 | 36 |
| (8960, 1536) | 16 | 16 | 20 | 20 |
| (896, 4864) | 8 | 8 | 8 | 8 |
| (4096, 1024) | 12 | 12 | 16 | 16 |
| (3584, 18944) | 8 | 8 | 8 | 8 |
| (4096, 4096) | 20 | 20 | 24 | 24 |
| (18944, 3584) | 32 | 32 | 36 | 36 |
| (1536, 1536) | 16 | 12 | 16 | 12 |
| (1536, 256) | 8 | 8 | 8 | 8 |
| (896, 128) | 8 | 8 | 8 | 8 |
| (14336, 4096) | 32 | 32 | 40 | 36 |
| (4864, 896) | 12 | 16 | 20 | 16 |
| (3584, 512) | 8 | 12 | 8 | 12 |
| (896, 896) | 8 | 8 | 8 | 8 |
| (4096, 14336) | 8 | 8 | 8 | 8 |
| (1536, 8960) | 8 | 8 | 8 | 8 |

# I   Full Evaluation Results on Jetson Orin AGX

Figure 14 presents full results on Jetson Orin AGX. Due to its powerful SSD and compute capability, overall latency is lower compared to Jetson Orin Nano.

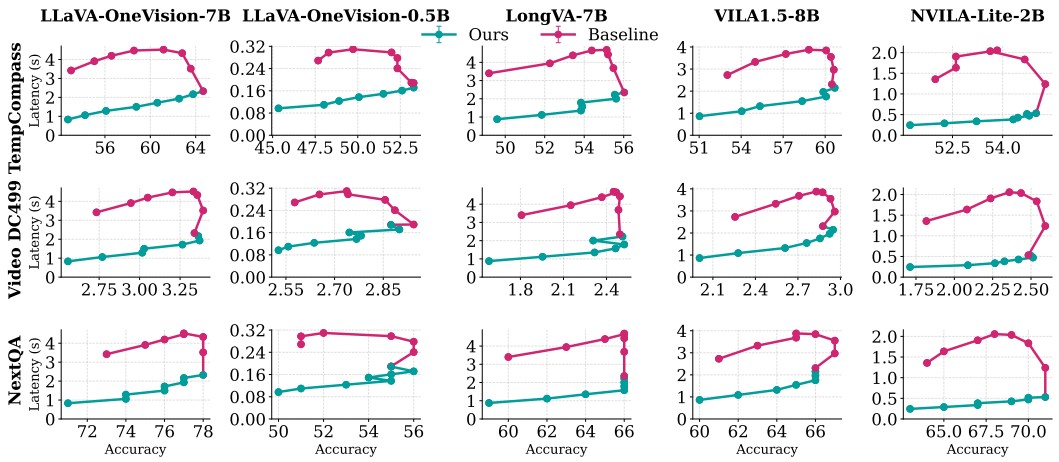

Figure 14: End-to-end performance on Jetson Orin AGX.

## J Extended Visualization of Mask Patterns and Contiguity

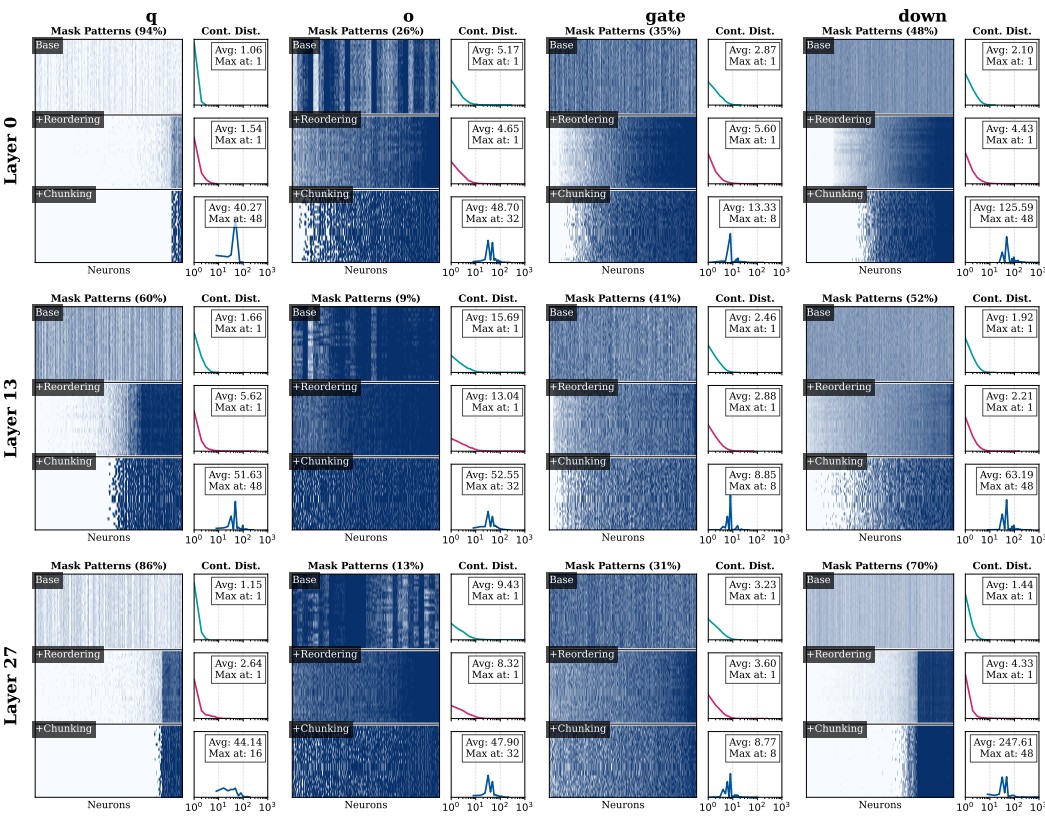

Figure 15: Mask patterns and corresponding contiguity distributions before and after applying our method, shown across different layers (0, 13, 27) and activation types (q, o, gate, down) at sparsity=0.4.

Figure 10 presented a case study on the effect of our method on mask patterns and contiguity distributions for layer 0, activation type q, under effective sparsity 0.3 and profiled sparsity 0.9.

Here, we provide an extended visualization across a broader range of settings in Figure 15. The visualization is structured as a $3 \times 4$ grid, where each row corresponds to a layer and each column

to an activation type. Each cell contains two subfigures: the left shows the binary mask patterns for three configurations—baseline, baseline with reordering, and baseline with reordering plus chunking—stacked vertically. The $x$-axis represents neuron index and the $y$-axis represents different input samples. The right subfigure presents the corresponding contiguity distribution, with the $x$-axis (log-scaled) denoting chunk size and the $y$-axis showing density. We annotate both the average and the mode (i.e., the most frequent chunk size) for each distribution.

These visualizations highlight that our method consistently promotes contiguity across layers and activation types. Qualitatively, the mask patterns become visibly less fragmented, particularly in high-sparsity regimes such as q and `down`. Quantitatively, both the average and mode of the contiguity distribution shift toward larger chunk sizes. While offline reordering provides marginal improvements, the majority of the contiguity gain arises from our online chunk selection policy, which adapts to input-dependent activation patterns (see Appendix F).

## K    Effect of Visual Token Density

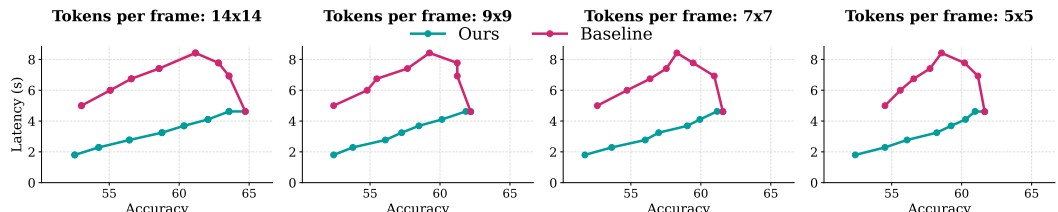

Figure 16: End-to-end performance on Jetson Orin Nano under different token counts per frame.

A large number of tokens per frame limits how many frames can fit within the model's context window. To address this, various token reduction techniques have been proposed, ranging from simple spatial pooling [55] to more advanced clustering [45]. We evaluate the impact of per-frame token count in Figure 16, using spatial pooling to control the number of tokens. As the token count decreases, we observe a modest drop in accuracy across all methods. Nonetheless, our method consistently outperforms the baseline, indicating that the I/O benefits of our approach are robust to changes in visual token density.

## L    Extended Comparison with Related Methods

We provide extended comparisons with three major classes of related methods: (i) *LLM in a Flash* [2], which also addresses flash-offloaded inference; (ii) *TEAL* [24], upon which our baseline implementation is built; and (iii) regularization-based pruning methods (e.g., L1 and group-Lasso), which promote sparsity through training-time penalties.

**LLM in a Flash.** *VLMs' smooth activation distributions require fundamentally different approaches.* Our work identifies critical inefficiencies in flash-offloaded VLM inference unaddressed by LLM in a Flash (hereafter LLMFlash). LLMFlash targets ReLU-based LLMs with >90% sparsity; we target non-ReLU VLMs with smoother activation distributions requiring 40-60% sparsity. This difference creates a counterintuitive phenomenon: in VLMs, higher sparsity can increase latency due to fragmentation-induced throughput degradation outweighing data savings. This workload difference motivates our methodological divergence. While LLMFlash simply reduces total I/O volume using sparsification, we explicitly model I/O latency through contiguity-aware cost modeling and jointly consider neuron importance and I/O efficiency during sparsification. Given these workload differences, we now analyze why LLMFlash's core techniques cannot effectively address our contiguity challenges.

*Neuron bundling proves insufficient without explicit contiguity optimization.* LLMFlash employs row–column bundling, grouping weights corresponding to the same activation (up-projection columns with down-projection rows). However, bundling alone is insufficient in our setting due to both hardware and methodological differences. Hardware-wise, LLMFlash was evaluated on MacBooks, where throughput saturates at chunk sizes <100 KB, whereas our Jetson devices require 236–348 KB for peak throughput (Figure 4a). This discrepancy likely arises from Jetson boards routing NVMe interrupts to a single CPU core, causing IOPS saturation, in contrast to MacBooks' multi-core

interrupt distribution [8, 42]. Methodologically, LLMFlash's up/down-projection bundling conflicts with our predictor-free approach, where each matrix is sparsified based on its own activations. Even when adapted to bundle matrices sharing input activations (e.g., Q/K/V or up/gate), the largest bundled weights (∼74 KB) achieve only half the optimal bandwidth on our hardware. Thus, bundling alone cannot achieve peak I/O performance—explicitly targeting contiguity as a design objective is essential.

Furthermore, when techniques such as TEAL's sparsification are combined with bundling, they introduce preprocessing and postprocessing overheads and can yield paradoxical I/O behavior. For instance, bundling Q/K/V matrices based on overlapping neurons improves locality for bundled weights but scatters remaining unbundled neurons across matrices, leading to fragmented reads. The net performance depends on whether contiguity gains outweigh fragmentation penalties, making bundling's effectiveness highly pattern-dependent.

Table 3: Comparison between our method and bundling-based implementations across models and datasets. Each cell shows two average speedup ratios: (1) ours vs. baseline and (2) ours vs. baseline+bundling.

| Dataset / Model | LLaVA-7B | LLaVA-0.5B | VILA-8B | NVILA-2B | LongVA |
|---|---|---|---|---|---|
| TempCompass | 2.06/2.41 | 2.05/1.94 | 1.60/1.83 | 3.24/3.76 | 2.15/2.50 |
| Video DC499 | 2.11/2.45 | 2.06/2.02 | 1.60/1.78 | 3.22/3.70 | 2.25/2.59 |
| NextQA | 1.76/1.98 | 2.12/1.99 | 1.50/1.70 | 3.44/3.96 | 2.04/2.34 |

Table 3 reports the experimental results comparing our method with bundling-based implementations. Our method achieves consistent speedups of 1.5–3.4× over the baseline and 1.7–4.0× over bundling-based implementations across all models and datasets. Bundling degrades performance in most cases except LLaVA-0.5B, confirming that its benefits are unpredictable and pattern-dependent, whereas our contiguity-aware approach consistently improves efficiency.

*Sliding window caching vs. offline reordering trades memory for adaptability.* LLMFlash's sliding-window caching maintains recently activated neurons' weights in memory, trading memory for reduced latency—an infeasible approach in our memory-constrained edge deployments. Both their caching and PowerInfer [38]'s hot neuron caching leverage additional memory to reduce flash accesses by exploiting statistical access patterns. Instead, we employ offline reordering for comparable benefits: both methods make frequently accessed weights cheaper to load, but caching is runtime-adaptive while consuming memory, whereas reordering imposes zero memory overhead while being less adaptive. In practice, we keep only essential weights—vision encoder, LM head, and KV cache—in device memory, representing the truly "hottest" components. Our chunk selection algorithm naturally accommodates any caching strategy by assigning zero importance to cached neurons, making it flexible and complementary to memory optimization approaches.

**TEAL.** Our method builds upon TEAL's fine-grained sparsity allocation across matrices rather than applying uniform sparsity. However, our focus differs fundamentally: TEAL determines *how much* to sparsify each layer under uniform access cost assumptions, whereas we determine *which* neurons to load at runtime by jointly considering activation importance and I/O efficiency in flash-based systems. This introduces latency-aware chunk selection that restructures selected neurons into contiguous memory layouts, aligning model-level sparsification with system-level latency behavior.

**Regularization-based pruning.** L1 regularization typically operates at the individual-weight level and rarely eliminates entire rows or columns of weights. As a result, it does not reduce the number of rows that must be loaded from flash, limiting its effect on activation sparsity and overall latency.

To meaningfully impact latency, sparsity must be *structured* (e.g., at the row or column level). This can be achieved by replacing L1 with group-Lasso regularization that applies L2-norm penalties to entire rows or columns. Column-wise regularization (e.g., applied to gate or up-projection matrices) encourages certain output activation channels to become zero, effectively deactivating the corresponding rows in the down-projection matrix and increasing activation sparsity. Row-wise regularization can also promote sparsity when the neuron-importance metric incorporates both activation magnitude and weight norm, lowering the importance of neurons associated with low-norm rows.

These regularization-based approaches are inherently input-agnostic: they prune the same weight regardless of the input context. This limits their achievable sparsity before severe accuracy degradation occurs—as shown in TEAL [30], where even 20% pruning causes noticeable performance drops.

## M   On Tradeoffs Between Accuracy and Latency

Our method is designed not to preserve accuracy at all costs, but to enable a more favorable tradeoff between accuracy and latency. In latency-sensitive deployments, it is often preferable to accept a modest accuracy degradation in exchange for significantly faster responses. The objective is not merely to match the performance of dense inference, but to shift the accuracy–latency Pareto frontier—achieving lower latency for comparable accuracy, or improved accuracy within a fixed latency budget. This tradeoff is particularly valuable in practical vision-language applications where the input is a video and the output is a natural language response. In many such scenarios, the user can quickly validate or refine the system's output, making responsiveness more critical than marginal gains in precision. Examples include:

- **Object or person retrieval.** When the user asks, e.g., "Where is the person in a red shirt?" or "Is the car still visible in this scene?", delivering fast candidate answers enables immediate visual verification and iteration.
- **Temporal localization.** In tasks like "When does the object fall?" or "At what time does the person enter the room?", coarse-grained temporal answers that arrive quickly are often more useful than delayed fine-grained ones.

In these streaming input scenarios, delayed responses can themselves degrade performance. This phenomenon—often referred to as *streaming accuracy*—has been observed in streaming perception literature, where the timeliness of model outputs directly influences their correctness [17, 19, 51, 52].

In such use cases, VLMs are part of interactive systems where user experience benefits more from fast responses than from exact answers. Our method supports this objective by enabling structured sparsification that reduces latency while maintaining actionable utility in response quality.

## N   Generalization to Other Use Cases

**Extension to multi-token LLM inference.** Although our method is evaluated in the context of vision-language models, the core idea—hardware-aware structured sparsification—extends beyond this domain. In particular, it is well-suited for multi-token LLM inference scenarios such as speculative decoding, parallel sampling for reasoning, and batched inference in interactive applications. In such settings, activations from multiple tokens are aggregated, leading to smoother neuron-importance distributions similar to those observed in VLMs. [5]

These multi-token inference workloads also underpin latency-critical user-facing systems such as chat assistants, copilots, and dialogue agents, which must maintain responsiveness under real-time constraints while often serving multiple concurrent requests. Minimizing end-to-end latency in these deployments is therefore essential, and extending our framework to such applications may offer broader benefits in practical LLM serving environments.

**Extension to plain LLM / ViT inference.** Our method can also be directly applied to plain LLMs and ViT-based models. The system relies on two key conditions: (i) the model exhibits smooth activation magnitude distributions (e.g., due to non-ReLU activations or multi-token inputs), and (ii) the hardware–model pair operates below I/O saturation when loading a single weight row. Recent LLMs increasingly employ smooth activation functions such as SwiGLU or GeLU, yielding moderate sparsity levels. While their activations are typically less smooth than those of multi-token scenarios, our method remains applicable, albeit with slightly smaller gains.

For ViT models, the transformer backbone remains largely compatible with our framework. As long as activation sparsity exists to a measurable degree, our approach can be applied without modification. Although ViTs are generally smaller (hundreds of millions of parameters), they still

---

[5]The sparsity mask generated from aggregated activations is shared across tokens, ensuring uniform inference latency across them (e.g., all samples in a batch finish simultaneously).

face I/O bottlenecks on resource-limited devices. In such cases, our chunking method becomes particularly valuable, as smaller weight channels make access fragmentation proportionally more severe.

To assess applicability to plain LLMs, we conducted a preliminary experiment on LLaMA3-8B and Qwen2-7B using the GSM8k dataset. We used the sum of selected neuron importance as a proxy for accuracy rather than full-dataset evaluation. We measured the importance–latency tradeoff in the first, middle, and last layers, observing average speedups of $1.22\times$ and $2.09\times$ for LLaMA3-8B and Qwen2-7B, respectively. These initial results suggest that our method generalizes to LLMs, though further work is needed to validate accuracy–latency tradeoffs at scale.

