# OpenReview forum: "VLM in a flash: I/O-Efficient Sparsification of Vision-Language Model via Neuron Chunking"
_NeurIPS.cc/2025/Conference — NeurIPS 2025 poster_

### Official Review · Reviewer_EnFY · 2025-06-22

**Clarity:** 3
**Significance:** 3
**Originality:** 3
**Rating:** 5
**Confidence:** 2

**Summary:**

This paper focuses on enhancing the efficiency of inference in vision-language models (VLMs) by addressing flash-based weight offloading and activation sparsification. It critically analyzes the current limitations of sparsification in VLMs and proposes a targeted approach that combines the importance of activation with the optimization of flash I/O efficiency.

**Questions:**

see weaknesses

**Ethical Concerns:**

["NO or VERY MINOR ethics concerns only"]

**Final Justification:**

My concerns are mostly addressed in the rebuttal.

**Limitations:**

yes

**Quality:**

3

**Strengths And Weaknesses:**

**Strengths**
* Innovative Approach: The introduction of a system-level approach that couples model sparsification with hardware I/O characteristics is a novel contribution. The paper emphasizes the importance of considering the underlying flash storage behavior, which is often overlooked in conventional sparsification methods.

* Comprehensive Evaluation: The paper presents detailed experiments on two different edge platforms, utilizing multiple models and benchmarks, and demonstrates consistent improvements across various setups.

**Weaknesses**

* In section 3.1, the comparison of VLM smoother distributions of activation magnitudes is based on a single model assumption （llava-onevision-7B）, and more models are needed for analysis and validation of the activation issue in VLMs.
* Limited Discussion on the profile cost and overheads in the latency Model. The latency model does not account for fixed overheads such as metadata lookup or command submission during flash read initiation. While this is mentioned in the paper, a more detailed discussion of how these factors could be incorporated in future work would help in improving the precision of the latency estimation, especially for smaller models or devices with limited throughput.

---

> ### Author Rebuttal · Authors · 2025-07-31
>
> We thank the reviewer for their thoughtful feedback and detailed evaluation of our work.
>
>
> ### Overview of the Review & Our Responses:
>
>
>
> * **S1 (Novel system-level approach):** We appreciate the reviewer's recognition of our innovative coupling of model sparsification with hardware I/O characteristics.
> * **S2 (Comprehensive evaluation):** We are glad the reviewer acknowledged our thorough experiments across multiple edge platforms, models, and benchmarks.
> * **W1 (Multi-model validation)**: We provide coefficient of variation analysis across 5 VLM models demonstrating universal smoothing phenomenon due to architectural properties, not model-specific behavior.
> * **W2 (Profiling costs and overheads)**: We clarify that profiling takes &lt;20 minutes per device and fixed overheads don't affect our algorithm's correctness due to its greedy ranking nature—designed to operate within strict 2ms runtime budgets. Future global optimization approaches requiring absolute latency precision would benefit from more precise latency modeling.
>
> We now provide detailed responses to each concern.
>
> *Note: The proposed additions will be included in the main paper, with supplementary material used if necessary due to space constraints.*
>
>
> ### W1: "In section 3.1, the comparison of VLM smoother distributions of activation magnitudes is based on a single model assumption (llava-onevision-7B), and more models are needed for analysis and validation of the activation issue in VLMs."
>
> The smoothing phenomenon is not model-specific but a mechanistic consequence of VLM architecture. This occurs due to two fundamental factors that apply universally to VLMs:
>
>
>
> * **Non-ReLU activation functions**: Modern VLMs universally adopt gated activations (SwiGLU, GeGLU) rather than ReLU, which produce continuous rather than sparse activations.
> * **Multi-token averaging**: VLMs process video frames as multiple patches (e.g., 14×14 = 196 tokens). Neuron importance is computed by averaging activation magnitudes across these tokens, which naturally reduces variation in importance scores.
>
> To empirically support this explanation, we measured the coefficient of variation (CV) of neuron importance before the down-projection layer—where conventional sparsification is typically applied in ReLU-based LLMs—ensuring consistency when comparing VLMs with a ReLU-based LLM (OPT-6.7B):
>
> | Layer  | LLaVA-7B | LLaVA-0.5B | VILA-8B | NVILA-2B | LongVA | OPT-6.7B |
> |--------|----------|------------|---------|----------|---------|----------|
> | First  | 1.44     | 1.31       | 1.25    | 1.07     | 1.20    | 11.65    |
> | Mid    | 1.25     | 1.33       | 1.38    | 1.32     | 1.34    | 8.63     |
> | Last   | 3.30     | 3.58       | 2.48    | 4.55     | 3.01    | 9.19     |
>
> All VLM models consistently show dramatically lower CV values (1.07-4.55) compared to the ReLU baseline (8.63-11.65), demonstrating this is an architectural property, not model-specific behavior. This smoothing makes contiguity-aware selection particularly beneficial for VLMs: when importance differences between neurons are small, I/O efficiency becomes the decisive factor for overall performance.
>
> We will enhance Section 3.1 to provide a more detailed explanation of why activation smoothness emerges in VLMs—specifically due to gated activation functions and multi-token averaging—and include the coefficient of variation measurements across multiple VLM architectures to demonstrate this is a universal architectural property rather than a model-specific phenomenon.
>
>
> ### W2: "Limited Discussion on the profile cost and overheads in the latency Model..."
>
> Our latency model design intentionally excludes fixed overheads without impacting algorithm effectiveness. Let us clarify both profiling costs and the rationale for this design choice:
>
> **Profiling Cost**: As noted in Line 179, the profiling process is detailed in Appendix B, where we measure throughput in 1KB increments until reaching 99% of peak performance, completing under 20 minutes per device. This one-time cost is negligible compared to deployment benefits.
>
> **Impact of Fixed Overhead and Greedy vs. Global Optimization Trade-offs**: Our greedy chunk selection uses importance-per-latency ratios for ranking candidates. Since fixed costs apply uniformly to all candidates, they don't affect the relative ranking used in our greedy selection. While global optimization algorithms would require absolute latency precision including fixed overheads, we use a greedy approach due to runtime constraints—executing ~196 times per frame for LLaVA-7B within a strict 2ms budget (Appendix H, Figure 4). As described in Lines 142-146, this frequent execution demands lightweight computation, making our current design well-suited for real-time inference.
>
> We will enhance Section 4.1.1 to discuss profiling overhead considerations, expand Section 4.1.2 to explain how fixed overheads interact with our greedy algorithm and why they don't affect relative ranking decisions, and add to Section 6 a discussion of how future global optimization approaches could benefit from more sophisticated latency models that account for absolute costs, including fixed overheads.

---

### Official Review · Reviewer_4gFS · 2025-06-29

**Clarity:** 2
**Significance:** 3
**Originality:** 3
**Rating:** 4
**Confidence:** 3

**Summary:**

This paper considers improving the latency of VLM inference on edge devices. Authors notice that while sparsely loading a fraction of neurons based on their importance can theoretically reduce the I/O overhead, this strategy ignores the flash storage behavior and in practice doesn’t lead to much latency gain. The authors propose to jointly consider the importance of neurons and the loading latency and select neurons by co-optimizing the two criteria. Experiments show that the authors’ method achieves lower latency compared to natively loading neurons solely based on their importance.

**Questions:**

- Please address the confusion as listed in the first point of the weaknesses.
- More comparisons with recent methods, including LLM in a Flash, would be helpful for better validating the effectiveness of the method.

**Ethical Concerns:**

["NO or VERY MINOR ethics concerns only"]

**Final Justification:**

Thank you for the detailed response. It has addressed all my questions and concerns. It would be great to include these clarifications and changes to the revised manuscript. I have adjusted my score accordingly.

**Limitations:**

Yes.

**Paper Formatting Concerns:**

No formatting concerns.

**Quality:**

2

**Strengths And Weaknesses:**

**Strengths:**

- The proposed co-optimization is intuitive and easy to understand, and the experimental results are promising for its effectiveness.
- The observation that relative ordering of latency is sufficient and crucial to the latency estimation is novel. In Figure 5, chunking-based latency estimation is more accurate than the conventional top-$k$ sparsification.
- Experiments demonstrate the effectiveness of the method on improving the latency compared to the native top-$k$ importance selection. The method achieves a large reduction of I/O latency in the breakdown.

**Weaknesses:**

- While the method is intuitively easy to understand, some parts of the description in the paper are confusing to me:
    - In Figure 1, how is the total latency $1.1$ calculated?
    - The chunking and reordering are not described with sufficient detail. Especially, the reordering is only described with several lines of text, but it is very important to the performance, as shown in the ablation study. More illustrations would be helpful.
    - In line 175, the authors mentioned that he selected rows are grouped into contiguous chunks. But if selected rows are like Row1, Row3, and Row5, how are they grouped in this case?
    - How is chunking designed to avoid overlapping from different scales? For example, in Figure 1 on the scale of two neurons, why is the group of 7&4 not selected?
- Lack of comparison to the recent method. While the authors mentioned LLM in a Flash and claimed it is suboptimal, I don’t find any comparison with it in the main body.

---

> ### Author Rebuttal · Authors · 2025-07-31
>
> We thank the reviewer for their thoughtful feedback and detailed evaluation of our work.
> ### Overview of the Review & Our Responses:
> * **S1-S3**: We acknowledge the reviewer's recognition of our co-optimization approach's intuitiveness, novel latency estimation insights, and experimental effectiveness.
> * **W1 (Technical clarity)**: We will revise technical descriptions to improve algorithmic clarity and reproducibility.
> * **W2 (Recent methods comparison)**: We position our work by analyzing how VLMs create fundamentally different I/O challenges than prior ReLU-based LLM, demonstrate why LLM in a Flash's techniques cannot address our problem, and clarify TEAL's orthogonal focus on sparsity profiling.
>
> We now provide detailed responses to each concern.
>
> *Note: The proposed additions will be included in the main paper, with supplementary material used if necessary due to space constraints.*
> ### W1: "While the method is intuitively easy to understand, some parts of the description in the paper are confusing..."
>
> We appreciate these clarification requests and address each technical detail below:
>
> **"In Figure 1, how is the total latency calculated?"**
>
> The total latency is the sum of individual chunk latencies, where each chunk's latency = data size ÷ throughput, with throughput values pre-profiled for each chunk size. For the example shown:
>
>
>
> * Chunk size 2 has throughput 4 → latency = 2/4 = 0.5
> * Chunk size 3 has throughput 5 → latency = 3/5 = 0.6
> * Total latency = 0.5 + 0.6 = 1.1
>
> We will add these formulas to Figure 1 to make the logic easy to follow.
>
> **"The chunking and reordering are not described with sufficient detail..."**
>
> **Chunking**: The complete algorithm with pseudocode is provided in Appendix C (Algorithm 1).
>
> We will make this reference more prominent and provide a more detailed description of algorithmic steps (detailing how we construct multi-scale chunk candidates, score them, and greedily select non-overlapping chunks) in Section 4.2.2 Algorithm Overview.
>
> **Reordering**: The full procedure involves three steps:
>
>
>
> 1. Count how often each neuron activates using calibration dataset
> 2. Sort neurons in decreasing frequency and permute weight matrix rows
> 3. At runtime, apply the same permutation to activation
>
> The runtime permutation operation incurs negligible overhead. For LLaVA-7B with shape (1, 196, 18944), we profiled this on Jetson Nano over 100 trials: mean overhead was 1.5ms with 95% confidence bound of 1.8ms, representing &lt;0.02% of total latency.
>
> We will add the above detailed explanations to Section 4.3 Additional Optimization: Hot-cold Reordering.
>
> **"If selected rows are like Row1, Row3, and Row5, how are they grouped?"**
>
> Only adjacent rows form contiguous chunks. In your example, rows 1, 3, and 5 would not be grouped; each would form a separate chunk. We group consecutive selected rows into larger chunks for basic I/O optimization—standard practice for improving memory access efficiency.
>
> We will revise Line 175 to: "Selected rows are grouped into sets of contiguous chunks, where only adjacent rows are merged into the same chunk."
>
> **"How is chunking designed to avoid overlapping from different scales?"**
>
> We generate multi-scale chunk candidates, sort by importance-per-latency ratio, then greedily select the highest-ranked non-overlapping chunks (Lines 231-232). In Figure 1, chunk 7&4 is excluded because the higher-ranked chunk 7&4&5 has already been selected.
>
> We will revise Lines 231-232 to: "Candidates are sorted by score, and the algorithm iteratively selects the highest-ranked chunks while excluding overlapping candidates until the budget is met."
>
>
> ### W2: "Lack of comparison to the recent method..."
>
> We provide comprehensive comparisons with the two most relevant recent methods: LLM in a Flash [1], which also addresses flash-offloaded inference, and TEAL [2], which our baseline directly builds upon.
>
> **LLM in a Flash**
>
> **VLMs' smooth activation distributions require fundamentally different approaches**: Our work identifies critical inefficiencies in flash-offloaded VLM inference unaddressed by LLM in a Flash (hereafter LLMFlash). LLMFlash targets ReLU-based LLMs with >90% sparsity; we target non-ReLU VLMs with smoother activation distributions requiring 40-60% sparsity. This difference creates a counterintuitive phenomenon: in VLMs, higher sparsity can increase latency due to fragmentation-induced throughput degradation outweighing data savings.
>
> This workload difference motivates our methodological divergence. While LLMFlash simply reduces total I/O volume using sparsification, we explicitly model I/O latency through contiguity-aware cost modeling and propose sparsification jointly considering neuron importance and I/O efficiency.
>
> Given these workload differences, we now analyze why LLMFlash's core techniques cannot effectively address our contiguity challenges.
>
> **Neuron bundling proves insufficient without explicit contiguity optimization:** LLMFlash uses row-column bundling, grouping weights corresponding to the same activation (up-projection columns with down-projection rows). However, bundling proves insufficient in our setting due to hardware and methodological differences. Hardware-wise, LLMFlash was evaluated on MacBooks where throughput plateaus at &lt;100KB, while our Jetson devices require 236-348KB for peak throughput (Figure 4a)—likely due to Jetson boards routing NVMe interrupts to single CPU cores causing IOPS saturation versus MacBooks' multi-core interrupt distribution [3, 4]. Methodologically, their up/down projection bundling conflicts with our predictor-free approach, where different matrices are sparsified based on respective input activations. Even when the method is adapted by bundling matrices sharing input activations (Q/K/V or up/gate), the largest bundled weights (~74KB) achieve only half the optimal bandwidth on our hardware. This demonstrates that bundling alone is insufficient—peak I/O performance requires explicitly targeting contiguity as a design objective.
>
> TEAL's sparsification introduces preprocessing/postprocessing overhead and creates paradoxical I/O effects when bundling is adopted. For example, when Q, K, V matrices are bundled based on overlapping neurons, bundled weights load efficiently but remaining unbundled neurons become scattered across matrices, forcing fragmented reads. The net performance depends on whether contiguity gains from bundling outweigh fragmentation penalties from unbundled neurons, making bundling unpredictably beneficial or detrimental depending on sparsity patterns.
>
> Our evaluation compares baseline sparsification with neuron bundling against our approach across five VLM models and three datasets:
>
> |Dataset/Model|LLaVA-7B|LLaVA-0.5B|VILA-8B|NVILA-2B|LongVA|
> |-|-|-|-|-|-|
> |**TempCompass**|2.06/2.41|2.05/1.94|1.60/1.83|3.24/3.76|2.15/2.50|
> |**Video DC499** |2.11/2.45|2.06/2.02|1.60/1.78|3.22/3.70|2.25/2.59|
> |**NextQA**|1.76/1.98|2.12/1.99|1.50/1.70|3.44/3.96|2.04/2.34|
>
> *Each cell shows two average speedup ratios: (1) our method vs. baseline, (2) our method vs. baseline with bundling*
>
> Our method achieves consistent average speedups of 1.5-3.4× over baseline and 1.7-4.0× over bundling implementations across all models and datasets. Bundling degrades performance in most cases except LLaVA-0.5B, confirming that bundling effects are unpredictable and pattern-dependent, while our contiguity-aware approach delivers reliable improvements.
>
> **Sliding window caching vs. offline reordering trades memory for adaptability:** LLMFlash's sliding window caching—maintaining recently activated neurons' weights in memory—represents a memory-for-performance tradeoff potentially infeasible in our memory-constrained edge deployment. Both their caching and PowerInfer's hot neuron caching [5] leverage additional memory to reduce flash accesses by exploiting statistical access patterns. Instead, we employ offline reordering for similar benefits: both techniques make frequently accessed weights cheaper to load, but caching is runtime-adaptive while consuming memory, whereas reordering imposes zero memory overhead while being less adaptive. We keep only essential weights in memory—vision encoder, LM head, and KV cache—accessed every inference iteration and representing truly "hottest" components. Our chunk selection naturally accommodates any caching strategy by assigning zero importance to cached neurons, making it flexible and complementary to memory optimization approaches.
>
> **TEAL**
>
> **How Much to Sparsify vs. Which Neurons to Select:** Our method builds upon TEAL's fine-grained sparsity allocation across weight matrices rather than uniform sparsity. However, our focus differs fundamentally: TEAL determines **how much to sparsify** each weight assuming uniform access cost, while we determine **which neurons to select **at runtime by jointly considering neuron importance and I/O efficiency on flash-based systems, introducing latency-aware chunk selection that restructures selected neurons into more contiguous memory layouts.
>
> To address these comparisons, we will restructure the paper: split the current Section 2 "Background and Related Work," with Section 2 focusing solely on "Background" and adding a new dedicated "Related Work" section before "Discussion and Future Work." This new section will provide detailed comparisons with LLMFlash and TEAL as outlined above, mapping the broader landscape of related approaches to clarify our method's positioning and contributions.
>
> [1] "Llm in a flash: Efficient large language model inference with limited memory." ACL’24 \
> [2]"Training-free activation sparsity in large language models." ICLR’25 \
> [3] "Optimizing storage performance with calibrated interrupts." TOS’22 \
> [4] NVIDIA Developer Forum. “Gen 3 PCIe NVMe SSD with x4 Lanes Gets Higher IOPS on Nano Compared to the Xavier NX.” 6 Sept. 2022 \
> [5] "Powerinfer: Fast large language model serving with a consumer-grade gpu." SOSP’24

---

> ### Comment · Reviewer_4gFS · 2025-08-05
>
> Thank you for the detailed response. It has addressed all my questions and concerns. It would be great to include these clarifications and changes to the revised manuscript. I have adjusted my score to 4.

---

> > ### Author Response · Authors · 2025-08-06
> >
> > Dear Reviewer,
> >
> > We are grateful for your positive response and confirmation that our rebuttal has successfully addressed the previous issues. The manuscript will be updated to reflect these discussions and your recommendations. We welcome any additional feedback you may have.
> >
> > Best regards,
> >
> > Authors

---

### Official Review · Reviewer_AaFb · 2025-07-03

**Clarity:** 3
**Significance:** 4
**Originality:** 3
**Rating:** 5
**Confidence:** 4

**Summary:**

Large VLMs exceed edge device memory, so weights are offloaded to external storage and loaded as needed, causing latency. The authors propose selection of contiguous, moderately important channels, rather than scattered, highly important ones, to improve memory access efficiency and reduces this latency.
Main contribution include I/O efficiency-aware activation sparsification approach and efficient chunk-based neuron selection algorithm for improving the efficiency in VLM inference.

**Questions:**

- What could be the effect of L1 regularization, which is known to encourage weight sparsity, on the accuracy-latency tradeoff? Can it be adapted to support chunk-level sparsity instead of neuron-level sparsity?
- Does this approach affect the efficiency of batch processing? For instance, there might be cases where different samples in a batch have different inference times after applying this method.
- It will be interesting to see how this method can be extended to plain LLMs and plain ViT-based models.

**Ethical Concerns:**

["NO or VERY MINOR ethics concerns only"]

**Final Justification:**

The authors have provided well-reasoned responses to the concerns raised in my initial review.
The paper addresses a practically important problem with hardware-aware approach. While related to structured pruning, the explicit modeling of I/O latency and chunk-based selection adds novelty. Given the quality of the work, its relevance to real-world deployment, and the comments of other reviewers, I am maintaining my rating.

**Limitations:**

yes

**Paper Formatting Concerns:**

No concerns.

**Quality:**

3

**Strengths And Weaknesses:**

- The paper presents a technically sound approach to making VLMs more efficient using hardware-aware sparsification. While the experimental evaluation is generally good, the paper would benefit from additional analysis on cases where the latency-accuracy tradeoff drops significantly.
- The manuscript is clearly written and logically organized.
- The paper addresses an important problem of translating model acceleration techniques to real-world latency gains on hardware.
- Although the proposed method is similar to structured pruning approaches, the incorporating I/O latency and approximating it using a model is novel.

---

> ### Author Rebuttal · Authors · 2025-07-31
>
> We thank the reviewer for their thoughtful feedback and detailed evaluation of our work.
>
> ### Overview of the Review & Our Responses:
>
> * **S&W1 (Latency–accuracy tradeoff behavior analysis):** We analyze the latency–accuracy curves in detail and explain the key factors that influence their behavior.
> * **S&W2 - S&W4 (Problem importance, Method novelty, Writing quality):** We appreciate the reviewer’s recognition of the technical soundness of our work and, in particular, the importance of bridging the gap between algorithmic optimizations and real-world latency—a direction we believe is critical for practical and deployable VLM systems.
> * **Q1 (Effect of L1 regularization):** We explain why standard L1 regularization does not directly benefit activation sparsity and propose structured regularization as a potential complementary approach.
> * **Q2 (Effect on batch processing):** We clarify our system’s behavior under batch processing, discuss its accuracy implications, and explain why our method becomes increasingly important.
> * **Q3 (Extension to LLMs and ViTs):** We confirm that our method applies to non-VLM models with preliminary quantitative results.
>
> We now provide detailed responses to each concern.
>
> *Note: The proposed additions will be included in the main paper, with supplementary material used if necessary due to space constraints.*
>
>
> ### W1: “the paper would benefit from additional analysis on cases where the latency-accuracy tradeoff drops significantly.”
>
> Since the reviewer’s comment on “cases where the latency–accuracy tradeoff drops significantly” may refer to different situations, we provide a comprehensive response covering multiple possible interpretations: (i) baseline latency increase at medium sparsity, (ii) accuracy increase in the low-sparsity regime despite higher sparsity, and (iii) potential limitations of our method under extreme conditions, such as scenarios involving severe sparsity or large memory budgets.
>
> (i) As noted in Figure 4b, naive magnitude-based sparsification can cause non-monotonic latency behavior, especially at medium sparsity levels, due to fragmented memory accesses. This results in counterintuitive regions in the baseline latency–accuracy curves (Figure 6) where latency increases even as more weights are sparsified. The effect is particularly pronounced for smaller models (LLaVA-0.5B, NVILA-2B), where each weight row is smaller and access fragmentation becomes more severe. In contrast, our method consistently yields monotonic latency behavior across models.
>
> (ii) A noticeable irregularity in the tradeoff curve appears in the low-sparsity regime, where we occasionally observe a slight increase in accuracy despite higher sparsity. We attribute this to a regularization effect from sparsification suppressing noisy activations, as stated in Line 281.
>
> (iii) At very high sparsity (e.g., >90% of ReLU-based LLMs), the opportunity for chunking diminishes since so few neurons remain active that there are insufficient contiguous regions to form meaningful chunks. However, such high sparsity often comes at the cost of significant accuracy degradation, which can limit the model's practical applicability.
>
> Alternatively, if the device has sufficient memory to cache a large portion of the model weights, the overall flash I/O volume becomes negligible, reducing the benefit of our method. Our work explicitly targets memory-constrained environments, where such caching is infeasible and flash I/O efficiency is critical for low latency.
>
> We will expand the Discussion section with a concise discussion clarifying the applicability boundaries of our method under extremely high sparsity or in memory-rich environments.
>
>
> ### Q1: “What could be the effect of L1 regularization, which is known to encourage weight sparsity, on the accuracy-latency tradeoff? Can it be adapted to support chunk-level sparsity instead of neuron-level sparsity?”
>
> We appreciate this insightful question, which prompted us to examine the impact of L1 regularization on our system more closely.
>
> L1 regularization operates at the individual weight level and rarely eliminates entire weight rows. Thus, it does not reduce the number of rows to be loaded, limiting its effect on activation sparsity and latency.
>
> To meaningfully impact latency, sparsity must be structured (e.g., at the weight row or column level). This can be achieved by replacing L1 with group Lasso regularization, which applies L2 norm penalties over entire rows or columns of weight matrices.
>
>
> * Column-wise regularization (e.g., on gate or up-projection matrix) would encourage certain output activation values to become zero. This, in turn, deactivates the corresponding rows in the down-projection matrix, increasing sparsity.
> * Row-wise regularization could also promote sparsity if the neuron importance metric incorporates weight norm in addition to the activation magnitude. This would lower the importance of the neurons that correspond to weight rows with low norm, increasing sparsity.
>
> The higher sparsity induced by these methods can potentially reduce the opportunity for chunking, thereby narrowing the performance gap between our method and the baseline.
>
> However, since these methods are input-agnostic, meaning they prune the same structure regardless of context. This limits their achievable sparsity before accuracy drops sharply—as shown in [2], where even 20% pruning causes noticeable degradation.
>
> We will add a discussion in the Related Works section introducing L1 regularization as a potential method to increase sparsity and explaining how structured variants such as group lasso could interact with our method, while also noting their practical limitations.
>
>
> ### Q2: “Does this approach affect the efficiency of batch processing? For instance, there might be cases where different samples in a batch have different inference times after applying this method.”
>
> We appreciate this question, as batch processing is an important consideration for real-world deployment.
>
> Our method does not negatively impact batch processing efficiency. In typical batched inference, all samples in a batch share the same model weights during computation. Thus, we apply a shared sparsity mask across the batch, loading the same subset of weights for every sample. This ensures uniform inference time across the batch.
>
> Interestingly, this shared-mask setting further smooths the neuron-importance distribution by aggregating importance across samples. This makes contiguity-aware sparsification even more critical, as adjacent neurons are more likely to have moderately high values, creating greater opportunities for chunking. However, sharing a single mask across a large batch may sharply degrade accuracy, requiring a lower sparsity level to maintain accuracy.
>
> That said, our target deployment scenario is memory-constrained edge devices, where single-sample inference is more common than large-batch processing. Nonetheless, our method remains compatible with both regimes and can provide benefits in batched settings as well.
>
> We will explain in the Discussion section how batching changes the sparsification behavior and the benefit of our chunking method under such settings.
>
>
> ### Q3: “It will be interesting to see how this method can be extended to plain LLMs and plain ViT-based models.”
>
> Our method can be directly applied to both LLMs and ViT models. Our system relies on two conditions: (i) models with smooth activation magnitude distributions (e.g., due to non-ReLU activation functions and multi-token inputs), and (ii) hardware paired with models small enough that reading a single weight row does not saturate I/O throughput.
>
>
>
> * Recent LLMs increasingly adopt smooth activation functions such as SwiGLU or GeLU, resulting in medium-level sparsity[3]. While the activation distribution may be less smooth than in multi-token VLMs, our method remains applicable, albeit with slightly reduced benefit.
> * Moreover, LLMs are often used with multi-token inputs in settings such as the verification stage of speculative decoding or tree-based parallel generation. These scenarios further smooth out the activation distribution, making them closely aligned with the VLM use cases we evaluated.
> * For ViT models, the transformer architecture is largely similar to that of LLMs and VLMs. As long as some degree of activation sparsity is present—which needs verification—our method can be applied without modification. That said, ViTs are typically smaller in size (on the order of hundreds of millions of parameters [4,5]), which reduces the overall demand for sparsification. However, on extremely resource-constrained devices where even ViTs cannot be fully cached in memory, the smaller per-row weight size makes our chunking method even more valuable, as the access fragmentation becomes more severe.
>
> We conducted a preliminary experiment to assess the applicability to LLMs (Llama3-8B, Qwen2-7B with GSM8k dataset). Due to time constraints, we used the sum of selected neuron importance as a proxy for accuracy instead of running full-dataset accuracy evaluations. We measured the importance–latency tradeoff in the first, middle, and last layers, observing speedup of 1.22x/2.09x on average for Llama3-8B, Qwen2-7B.
>
> We will update Section 5 to explicitly demonstrate that our approach is applicable beyond VLMs by including these quantitative results, and we will add further detailed experiments in the revision to strengthen this analysis.
>
> [1] "Learning structured sparsity in deep neural networks." NIPS’16
>
> [2] "Llm-pruner: On the structural pruning of large language models." NeurIPS’23
>
> [3] "Training-free activation sparsity in large language models." ICLR’25
>
> [4] "Learning transferable visual models from natural language supervision." International conference on machine learning. ICML’21.
>
> [5] "An image is worth 16x16 words: Transformers for image recognition at scale." ICLR’21

---

> > ### Comment · Reviewer_AaFb · 2025-08-06
> >
> > Thank you for the responses. The additional analysis on latency–accuracy tradeoff behavior, structured regularization, and batch processing clarified key aspects of your method and its practical implications. I also appreciate the preliminary results showing applicability to LLMs and ViTs.
> >
> > Overall, your rebuttal effectively addressed my concerns and strengthened my understanding of the contributions.

---

### Official Review · Reviewer_3ayJ · 2025-07-04

**Clarity:** 3
**Significance:** 3
**Originality:** 3
**Rating:** 4
**Confidence:** 3

**Summary:**

This paper proposes ​​an activation sparsification method named Neuron Chunking​​, which is designed to accelerate vision-language models on edge devices. The method improves flash-based weight offloading with a contiguity-based latency model to predict the cost of access patterns and a multi-scale chunk selection algorithm to explicitly consider actual I/O latency by selecting contiguous chunks. The method is verified based on various VLMs on edge devices, including Jetson Orin Nano and Jetson AGX Orin.

**Questions:**

Please refer to my comments above.

**Ethical Concerns:**

["NO or VERY MINOR ethics concerns only"]

**Final Justification:**

Thanks for the feedback. My concerns are mostly addressed. After reading the rebuttal as well as other reviews, I would like to keep my initial rating and lean towards accepting this paper.

**Limitations:**

Yes

**Quality:**

3

**Strengths And Weaknesses:**

Strengths:

- The paper is well-written with a clearly stated motivation and a reasonable and easy-to-understand method.

- Extensive studies are conducted to show the access patterns and model performance. The proposed method achieves decent improvements over the baseline activation sparsification method.

Weaknesses:

- The method proposed in the paper is largely based on existing work on acceleration for flash-offloaded LLM inference. The idea of latency analysis is closely related to the observation in [2]. The paper can be viewed as an extension of [2] for VLM acceleration.

- The paper focuses on studying the problem of accelerating video understanding inference, where a frame appending stage is considered between the prefill and decoding stage in inference, which I believe may not be the most common case for VLM inference. Will the observation (access pattern, speed-up results) be the same if we consider a simple image understanding task? The method can be more general if the solution is also valid for general vision-language understanding scenarios.

---

> ### Author Rebuttal · Authors · 2025-07-31
>
> We thank the reviewer for their thoughtful feedback and detailed evaluation of our work.
>
>
> ### Overview of the Review & Our Responses:
>
>
>
> * **S1 (Clear motivation and method):** We appreciate the reviewer’s positive feedback on the clarity and presentation of our paper.
> * **S2 (Extensive evaluation):** We are glad the reviewer recognized our extensive experimental study, which demonstrates substantial speedups on real hardware and consistent improvements across diverse models.
> * **W1 (Distinction from LLM in a Flash):** We position our work by analyzing how VLMs create fundamentally different I/O challenges than prior ReLU-based LLMs and demonstrate why LLM in a Flash's techniques cannot address our problem.
> * **W2 (VLM scenario generality):** We highlight the growing importance of online video understanding for practical applications with supporting references. We also explain that our system’s underlying conditions hold for single-image tasks, though with smaller end-to-end impact.
>
> We now provide detailed responses to each concern.
>
> *Note: The proposed additions will be included in the main paper, with supplementary material used if necessary due to space constraints.*
>
>
> ### W1: “The paper can be viewed as an extension of LLM in a Flash[1] for VLM acceleration.”
>
> While both our work and LLM in a Flash(hereafter LLMFlash) address I/O bottlenecks in flash-offloaded inference, our key contributions and technical focus diverge significantly:
>
> **VLMs' smooth activation distributions require fundamentally different approaches** : Our work identifies critical inefficiencies in flash-offloaded VLM inference unaddressed by LLM in a Flash (hereafter LLMFlash). LLMFlash targets ReLU-based LLMs with >90% sparsity; we target non-ReLU VLMs with smoother activation distributions requiring 40-60% sparsity. This difference creates a counterintuitive phenomenon: in VLMs, higher sparsity can increase latency due to fragmentation-induced throughput degradation outweighing data savings.
>
> This workload difference motivates our methodological divergence. While LLMFlash simply reduces total I/O volume using sparsification, we explicitly model I/O latency through contiguity-aware cost modeling and propose sparsification jointly considering neuron importance and I/O efficiency.
>
> Given these workload differences, we now analyze why LLMFlash's core techniques cannot effectively address our contiguity challenges.
>
> **Neuron bundling proves insufficient without explicit contiguity optimization:** LLMFlash uses row-column bundling, grouping weights corresponding to the same activation (up-projection columns with down-projection rows). However, bundling proves insufficient in our setting due to hardware and methodological differences. Hardware-wise, LLMFlash was evaluated on MacBooks where throughput plateaus at &lt;100KB, while our Jetson devices require 236-348KB for peak throughput (Figure 4a)—likely due to Jetson boards routing NVMe interrupts to single CPU cores causing IOPS saturation versus MacBooks' multi-core interrupt distribution [6, 7]. Methodologically, their up/down projection bundling conflicts with our predictor-free approach, where different matrices are sparsified based on respective input activations. Even when the method is adapted by bundling matrices sharing input activations (Q/K/V or up/gate), the largest bundled weights (~74KB) achieve only half the optimal bandwidth on our hardware. This demonstrates that bundling alone is insufficient—peak I/O performance requires explicitly targeting contiguity as a design objective.
>
> TEAL's sparsification [2] introduces preprocessing/postprocessing overhead and creates paradoxical I/O effects when bundling is adopted. For example, when Q, K, V matrices are bundled based on overlapping neurons, bundled weights load efficiently but remaining unbundled neurons become scattered across matrices, forcing fragmented reads. The net performance depends on whether contiguity gains from bundling outweigh fragmentation penalties from unbundled neurons, making bundling unpredictably beneficial or detrimental depending on sparsity patterns.
>
> Our evaluation compares baseline sparsification with neuron bundling against our approach across five VLM models and three datasets:
>
> |Dataset/Model|LLaVA-7B|LLaVA-0.5B|VILA-8B|NVILA-2B|LongVA|
> |-|-|-|-|-|-|
> |**TempCompass**|2.06/2.41|2.05/1.94|1.60/1.83|3.24/3.76|2.15/2.50|
> |**Video DC499** |2.11/2.45|2.06/2.02|1.60/1.78|3.22/3.70|2.25/2.59|
> |**NextQA**|1.76/1.98|2.12/1.99|1.50/1.70|3.44/3.96|2.04/2.34|
>
> *Each cell shows two average speedup ratios: (1) our method vs. baseline, (2) our method vs. baseline with bundling*
>
> Our method achieves consistent average speedups of 1.5-3.4× over baseline and 1.7-4.0× over bundling implementations across all models and datasets. Bundling degrades performance in most cases except LLaVA-0.5B, confirming that bundling effects are unpredictable and pattern-dependent, while our contiguity-aware approach delivers reliable improvements.
>
> **Sliding window caching vs. offline reordering trades memory for adaptability:** LLMFlash's sliding window caching—maintaining recently activated neurons' weights in memory—represents a memory-for-performance tradeoff potentially infeasible in our memory-constrained edge deployment. Both their caching and PowerInfer's hot neuron caching [8] leverage additional memory to reduce flash accesses by exploiting statistical access patterns. Instead, we employ offline reordering for similar benefits: both techniques make frequently accessed weights cheaper to load, but caching is runtime-adaptive while consuming memory, whereas reordering imposes zero memory overhead while being less adaptive. We keep only essential weights in memory—vision encoder, LM head, and KV cache—accessed every inference iteration and representing truly "hottest" components. Our chunk selection naturally accommodates any caching strategy by assigning zero importance to cached neurons, making it flexible and complementary to memory optimization approaches.
>
> To address these comparisons, we will restructure the paper: split the current Section 2 "Background and Related Work," with Section 2 focusing solely on "Background" and adding a new dedicated "Related Work" section before "Discussion and Future Work." This new section will provide detailed comparisons with LLMFlash as outlined above, mapping the broader landscape of related approaches to clarify our method's positioning and contributions.
>
>
> ### W2: “The paper focuses on studying the problem of accelerating video understanding inference ... Will the observation (access pattern, speed-up results) be the same if we consider a simple image understanding task?”
>
> While early VLMs primarily focused on single-image understanding, the field is rapidly shifting toward online video understanding[3, 4, 5], enabled by models trained on large-scale video datasets. This transition is especially critical for enabling practical, real-world systems such as AR assistants with smart glasses and autonomous agents like robots. In these scenarios, the model continuously processes incoming video frames, with user queries arriving intermittently. As a result, the frame appending phase dominates inference time, while prefill (initial prompt setup) and decoding (answer generation) occur less frequently. We formally described this inference pipeline in Appendix D.1.
>
> Regarding single-image understanding tasks, the core conditions that our system targets—namely, (i) models with smooth activation magnitude distributions (e.g., due to non-ReLU activation functions and multi-token inputs), and (ii) hardware paired with models small enough that reading a single weight row does not saturate I/O throughput—remain the same. Therefore, the speed-up trends we observe should hold similarly in that setting. However, in single-image tasks, the total latency is often dominated by decoding rather than image processing. Thus, while our chunking method may still reduce image processing latency, its relative impact on end-to-end latency may be less significant.
>
> We will revise the Introduction and Related Works sections to explicitly define our target scenario and emphasize its importance for practical real-world applications. We will also add a brief discussion in the Discussion section clarifying how our system’s benefits translate to single-image tasks.
>
> [1] "Llm in a flash: Efficient large language model inference with limited memory." ACL’24.
>
> [2] "Training-free activation sparsity in large language models."ICLR’25.
>
> [3] "Videollm-online: Online video large language model for streaming video." CVPR’24.
>
> [4] "OVO-Bench: How Far is Your Video-LLMs from Real-World Online Video Understanding?." CVPR’25.
>
> [5] ChatGPT voice mode. [https://help.openai.com/en/articles/8400625-voice-mode-faq#h_fa3ca78ac7](https://help.openai.com/en/articles/8400625-voice-mode-faq#h_fa3ca78ac7)
>
> [6] "Optimizing storage performance with calibrated interrupts." TOS’22 \
>
> [7] NVIDIA Developer Forum. “Gen 3 PCIe NVMe SSD with x4 Lanes Gets Higher IOPS on Nano Compared to the Xavier NX.” 6 Sept. 2022
>
> [8] "Powerinfer: Fast large language model serving with a consumer-grade gpu." SOSP’24

---

> > ### Author Response · Authors · 2025-08-08
> >
> > Dear reviewer,
> >
> > Thank you for your time and consideration. As the discussion period is approaching its end, we wanted to kindly check in regarding our rebuttal response. If you have had a chance to review it, we would greatly appreciate your feedback. Please let us know if there is anything we can clarify or if you need additional information.
> >
> > Best regards,
> >
> > Authors

---

### Decision · Program_Chairs · 2025-09-17

**Decision:**

Accept (poster)

**Comment:**

This paper proposes an efficiency-aware activation sparsification approach (Neuron Chunking) designed to accelerate vision-language models on edge devices. This is done through an efficient chunk-based neuron selection algorithm for improving the efficiency in VLM inference. While the proposed method is largely based on existing work on acceleration for flash-offloaded LLM inference, reviewers agree that this is a technically sound approach to making VLMs more efficient using hardware-aware sparsification. The paper is well written and the approach is solid and well validated empirically.